# Enhancing and inhibitory motifs regulate CD4 activity

Mark S Lee[1], Peter J Tuohy[1], Caleb Y Kim[1], Katrina Lichauco[1], Heather L Parrish[1], Koenraad Van Doorslaer[1,2,3,4,5]*, Michael S Kuhns[1,3,4,5,6]*

[1]Department of Immunobiology, The University of Arizona College of Medicine, Tucson, United States; [2]School of Animal and Comparative Biomedical Sciences, University of Arizona, Tucson, United States; [3]Cancer Biology Graduate Interdisciplinary Program and Genetics Graduate Interdisciplinary Program, The University of Arizona, Tucson, United States; [4]The BIO-5 Institute, The University of Arizona, Tucson, United States; [5]The University of Arizona Cancer Center, Tucson, United States; [6]The Arizona Center on Aging, The University of Arizona College of Medicine, Tucson, United States

**Abstract** CD4+ T cells use T cell receptor (TCR)–CD3 complexes, and CD4, to respond to peptide antigens within MHCII molecules (pMHCII). We report here that, through ~435 million years of evolution in jawed vertebrates, purifying selection has shaped motifs in the extracellular, transmembrane, and intracellular domains of eutherian CD4 that enhance pMHCII responses, and covary with residues in an intracellular motif that inhibits responses. Importantly, while CD4 interactions with the Src kinase, Lck, are viewed as key to pMHCII responses, our data indicate that CD4–Lck interactions derive their importance from the counterbalancing activity of the inhibitory motif, as well as motifs that direct CD4–Lck pairs to specific membrane compartments. These results have implications for the evolution and function of complex transmembrane receptors and for biomimetic engineering.

*For correspondence:
vandoorslaer@arizona.edu (KVD);
mkuhns@email.arizona.edu (MSK)

## Editor's evaluation

This paper takes an evolutionary approach to investigate the mechanisms by which CD4 regulates T-cell receptor activation and downstream functional responses. The authors identify conserved and coevolving motifs in the extracellular, transmembrane, and intracellular domains of CD4 that appear to regulate multiple aspects of its function. These findings suggest a recalibration of the perception of CD4 as simply an accessory to the central complex of T-cell receptors and CD3 in pMHC-specific T-cell responses.

## Introduction

The immunological 'Big Bang' that gave rise to RAG-based antigen receptor gene rearrangement in jawed vertebrates produced an adaptive immune system in which each naive B and T cell expresses a clonotypic B or T cell receptor (BCR or TCR) with unique antigen specificity (**Bernstein et al., 1996**; **Flajnik, 2014**). The B and T cell repertoires can therefore be thought of as combinatorial libraries from which individual clonotypes, expressing receptors specific to antigen, expand to mount a tailored response. This strategy provides jawed vertebrates with long-lived protection against microbial infection, neoplastic transformation, and is the basis for vaccines; yet it also presents the risk of reactivity against self. As a result, mechanisms have evolved to ensure that the adaptive immune system of jawed vertebrates is on high alert to respond to foreign antigens while maintaining tolerance to self.

For the CD4$^+$ T cell repertoire, discriminating self from foreign begins in the thymus where developmental checkpoints test the strength with which a thymocyte's clonotypic TCR interacts with composite surfaces of self-peptides embedded within class II MHC (pMHCII) (*Huseby et al., 2005*). TCRs that interact weakly with self-pMHCII direct positive selection toward the CD4$^+$ lineage, while those that strongly recognize self-pMHCII induce apoptosis to establish central tolerance by removing autoreactive TCR clonotypes from the repertoire by negative selection. For CD4$^+$ T cells that emerge from the thymus, the nature of TCR–pMHCII engagement determines their homeostasis, activation, and differentiation to one of a variety of helper (Th) or regulatory (Treg) phenotypes (*Gottschalk et al., 2010*; *Tubo and Jenkins, 2014*). These Th and Treg cells then influence the responses of a variety of other immune cell types.

The conversion of pMHCII-specific information into intracellular signals is an emergent property of five distinct modules: TCR, CD3γε, CD3δε, CD3 ζ ζ , and CD4 (*Kobayashi et al., 2020*; *Kuhns and Davis, 2012*). The TCR is the receptor module. It deciphers information encoded within the composite pMHCII surface and relays the information to the immunoreceptor tyrosine-based activation motifs (ITAMs) of three associated signaling modules (CD3γε, CD3δε, and CD3 ζ ζ ) (*Chen et al., 2022*; *Gil et al., 2002*; *Lee et al., 2015*). CD4 is the coreceptor module. It binds MHCII on the outside of the CD4$^+$ T cell and interacts with the Src kinase, Lck, via an intracellular CQC clasp motif (*Kim et al., 2003*; *Turner et al., 1990*). According to the TCR signaling paradigm, CD4 recruits Lck to phosphorylate the ITAMs of TCR–CD3 complexes when both TCR–CD3 and CD4 coincidently engage pMHCII (*Rudd, 2021*). Phosphorylation of the ITAMs initiates pMHCII-specific signaling, connecting TCR–CD3 complexes to the broader intracellular signaling machinery (*Courtney et al., 2018*; *Gaud et al., 2018*). In this model, the biophysical properties that govern TCR–pMHCII interactions are the key determinants for T cell fate decisions while CD4 plays a supporting role.

Recent evidence suggests however that the role of CD4 within the TCR signaling paradigm requires refinement. CD4's extracellular domain (ECD) can increase TCR dwell time on pMHCII and position its intracellular domain (ICD) in a defined relationship with the TCR–CD3 complex through coordinated rather than coincident interactions (*Glassman et al., 2016*; *Glassman et al., 2018*; *Guy and Vignali, 2009*). Furthermore, CD4 molecules that are not associated with Lck have been proposed to compete with those that are to limit the number of TCR–CD3 complexes phosphorylated by Lck, thus setting a threshold for the duration of TCR–pMHCII interactions required to initiate signaling (*Stepanek et al., 2014*). In addition, CD4 molecules that are associated with Lck are reported to play a vital role in pMHCII restriction by sequestering Lck away from TCR–CD3 to prevent off-target signaling by TCR interactions with non-pMHCII molecules (*Van Laethem et al., 2013*). These models help explain how the stability and composition of TCR–CD3–pMHCII–CD4(+/−Lck) assemblies influence and regulate ITAM phosphorylation. They also suggest that CD4, which has been less-well studied than the TCR, warrants more attention.

Accordingly, we reconstructed the evolutionary history of extant CD4 homologs from boney fish, reptiles, birds, and mammals to evaluate the results of ~435 million years of CD4 evolution in jawed vertebrates. Our analyses identified five putative motifs within the CD4 transmembrane domain (TMD) and ICD that are unique to mammals, or found only in eutherians (placental mammals), and contain residues under purifying selection. Further analyses identified residues within motifs in the ECD, TMD, and ICD that have covaried over evolutionary time. Follow-on structure–function analyses revealed a paradox that cannot be explained by the current TCR signaling paradigm. Specifically, mutating the transmembrane and intracellular motifs increased CD4–Lck association and impaired CD4-driven responses. Conversely, mutating the ICD helix or a motif therein reduced CD4–Lck interactions and enhanced responses. These findings have broad implications for how multisubunit transmembrane receptors relay ligand-specific information across the cell membrane, for our understanding of CD4$^+$ T cell biology, and for biomimetic engineering of synthetic receptors.

## Results

### Evolutionary analysis of CD4

We performed multiple analyses of available vertebrate CD4 ortholog sequences ($n$ = 99 distinct sequences), representing ~435 million years of evolution, to understand how ancient and ongoing environmental challenges have influenced CD4. The analyzed sequences represent fish, reptiles (including

birds), marsupials, and placental mammals. Details related to ortholog selection are outlined in Materials and methods. All sequences and files are available through the DataDryad repository associated with this manuscript. We used mouse CD4 (numbering by UniProt convention) as a reference to facilitate comparisons between evolutionary analyses and experimental studies.

Analysis of sequence conservation between the full set of extant CD4 molecules, or mammalian CD4 molecules only, showed particular conservation in the ICD (*Figure 1—figure supplement 1A, B*). To investigate the type of evolutionary selection shaping CD4 evolution, we determined nonsynonymous (dN) and synonymous (dS) substitution rates. Codons under diversifying selection have a dN:dS ratio >1 and those under purifying selection have a dN:dS ratio <1 (*Figure 1—figure supplement 1C, D*). The codon-specific dN:dS ratios were calculated using a fixed effects likelihood (FEL) method on both the full and mammalian only datasets (*Kosakovsky Pond and Frost, 2005*). Of the 17 codons under diversifying selection, 16 (94.1%) are distributed across the CD4 ectodomain while only one is found in the TMD. Of the 126 residues under purifying selection, 98 are distributed across the CD4 ectodomain (24.8% of all codons in the ECD). In contrast, 45.5% of TMD codons (10 of 22) and 45% (18 of 40) within the ICD are under purifying selection. These data suggest that mutating putative linear motifs within the ICD is selected against, arguing that more than just the CQC clasp is important for CD4 function (*Babu et al., 2011*; *Capra and Singh, 2007*; *Dyson and Wright, 2005*; *Gibson, 2009*; *Kim et al., 2003*; *Tompa, 2011*).

To further characterize the evolution of these motifs, we generated a maximum likelihood phylogenetic tree and predicted most recent common ancestor sequences at each node (*Figure 1A* and *Figure 1—figure supplement 1E*; *Hochberg and Thornton, 2017*). The conservation of specific residues in eutherian CD4 proteins is visualized using logo plot analysis to better consider positional variability of residues within this clade. Additionally, we provide a more complete picture of the pressures shaping CD4 molecules by associating the evolutionary selection signature (i.e., dN:dS ratio) with specific codons using both the mammalian only dataset, as well as all extant CD4 molecules (*Figure 1B* and *Figure 1—figure supplement 1E*). Particular attention was given to mouse and human CD4 due to their experimental and human health relevance.

First, we asked if our analyses would highlight residues that we know to be important for CD4 function by focusing on the evolutionary history of residues in the D3 domain of the CD4 ectodomain that form a solvent-exposed nonpolar patch in 3D space, stabilize TCR–CD3–pMHCII–CD4 assemblies, and increase pMHCII-specific responses (P228, F231, and P281) (*Figure 1C*; *Glassman et al., 2018*; *Wu et al., 1997*). The predicted most recent common ancestor of all amniotes contains a PLXF motif (mouse 228–231) in the D3 domain that is maintained in mammals (*Figure 1A*, node 1). P281 is not found in the predicted amniote most recent common ancestor but is in the mammalian most recent common ancestor, and extant mammals, suggesting that it arose after mammals diverged from reptiles. Importantly, P228, L229, F231, and P281 have small dN:dS values that are primarily driven by low dN rates, indicating that changing these residues likely affects fitness. Structural analysis indicates that L229 is buried in the hydrophobic core of the D3 domain as is L282 adjacent to P281, while the solvent-exposed P228, F231, and P281 impact CD4 function (*Glassman et al., 2018*; *Wu et al., 1997*). These analyses show that our approach identified D3 residues of known functional importance.

We next turned our attention toward the TMD and ICD by focusing on motifs in eutherian CD4 that: (1) deviate from ancestral sequences as well as those of other clades; (2) contain highly conserved residues with evidence of purifying selection. For example, the predicted most recent common ancestor of all amniotes (node 1) contains a GG patch (G402 and G403) in its TMD that is lost in nonavian reptiles but present in extant birds (*Figure 1A*). In the predicted eutherian most recent common ancestor (node 4), the GG patch expanded into a highly conserved GGXXG motif that is present in the majority of eutherian CD4 proteins (see logo plot) including mouse and human. Importantly, G402 and G403 were found to be under purifying selection using both the full and mammal-only dataset, while G406 was only identified in the full dataset (*Figure 1B* and *Figure 1—figure supplement 1E*). Our decision tree therefore identified this motif as potentially important for CD4 function. When considered with our previous work, and work on GXXXG motifs more broadly, the GGXXG motif is likely to mediate heterotypic protein–protein or protein–cholesterol interactions (*Parrish et al., 2015*; *Teese and Langosch, 2015*).

Our approach also identified a CV +C motif ('+' represents a basic residue) that is highly conserved in eutherian CD4 molecules but not present in the predicted mammalian most recent common ancestor

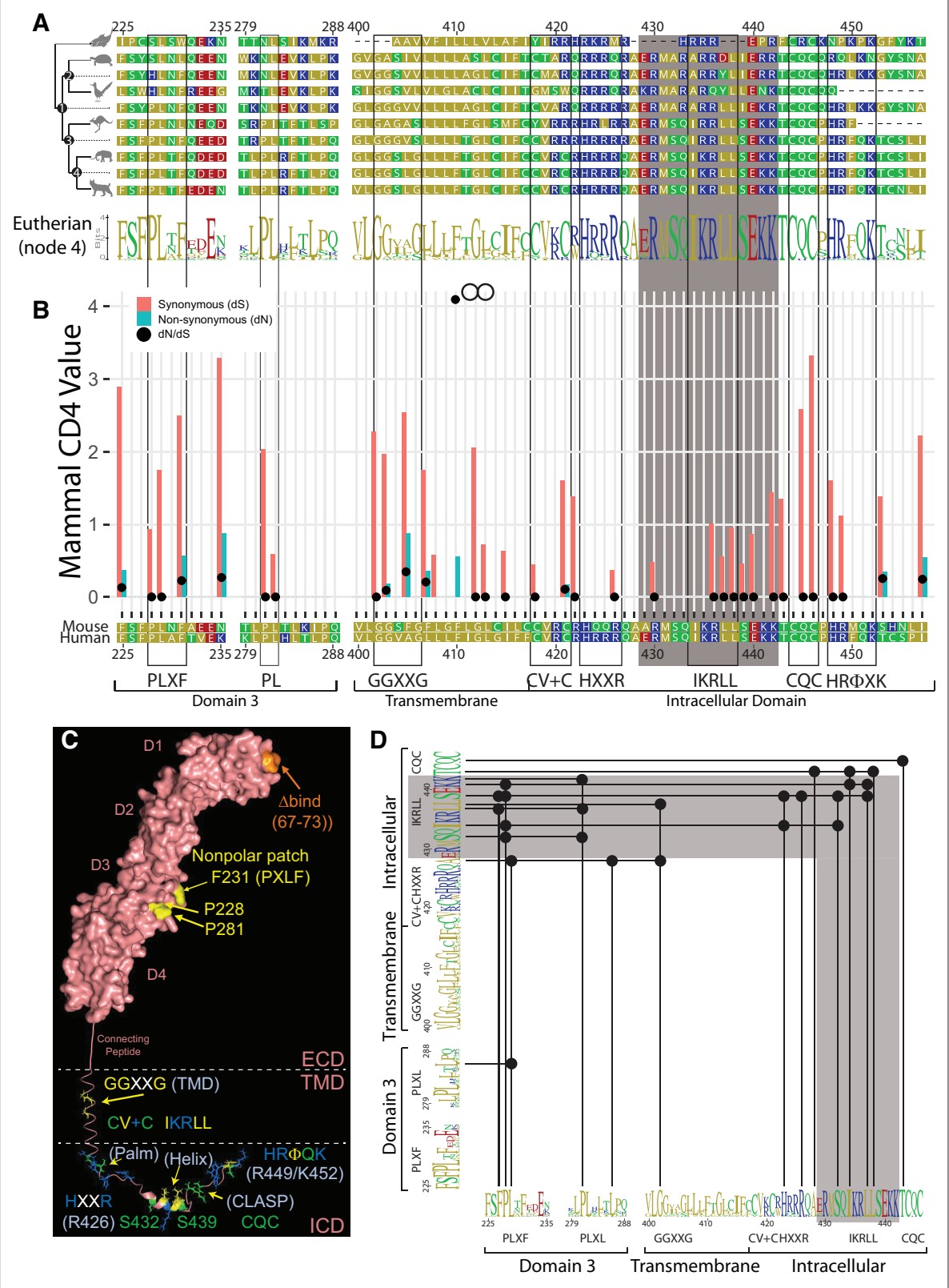

**Figure 1.** Evolutionary analysis of the CD4 molecule. (**A**) Reduced representation maximum likelihood phylogenetic tree clusters of CD4 sequences are shown with mouse CD4 numbering (uniprot) used as a reference. Residues are colored based on sidechain polarity. Dashes (-) indicate an evolutionary insertion or deletion event. Predicted most recent common ancestor (MRCA) sequences are shown at each node in the tree (Node 1-4). Logo plots of extant eutherian CD4 sequences are aligned at the bottom of the tree. Each stack of letters represents the sequence conservation at that position in

*Figure 1 continued on next page*

*Figure 1 continued*

the alignment. The height of symbols indicates the relative frequency of each amino acid at that specific position.(**B**) Synonymous (dS, red bars) and nonsynonymous (dN, blue bars) substitution rates within the CD4 coding sequence are shown as calculated for all CD4 orthologs included in the initial phylogenetic analysis using the Fixed Effects Likelihood (FEL) method. Only bars for which the likelihood ratio test indicated statistical significance (alpha = 0.1) are shown. Black circles show the ratio of both these values. Codons under diversifying selection have a dN:dS ratio >1. Those under purifying selection have a dN:dS ratio <1.(**C**) A theoretical structural model to show the relative location of the motifs discussed here. The surface rendered ECD of human CD4 (pdb 1WIQ) was joined with a connecting peptide and TMD (built using the PyMol Molecular Graphics system), and ICD (pdb 1Q68). Note here that mouse residue numbering (uniport) is used in this model for consistency with panels A–C.(**D**) Covarying residues were calculated using MISTIC2. Residues that covary are indicated with a black dot and connected with a solid line. Motifs identified in this study are indicated. The logo plot represents eutherian sequences. The complete MISTIC2 results matrix is available on Dryad (https://doi.org/10.5061/dryad.59zw3r26z). Boxes are used to highlight motifs discussed in this study, while the grey shading indicates the helix-turn region within the ICD. Key: MRCA = Most Recent Common Ancestor; FEL = Fixed Effects Likelihood; dS = Synonymous; dN = nonsynonymous; ECD = Extracellular Domain; TMD = Transmembrane Domain; ICD = Intracellular Domain.

The online version of this article includes the following figure supplement(s) for figure 1:

**Figure supplement 1.** Additional evolutionary analysis of the CD4 molecule.

or marsupials (*Figure 1A*). Palmitoylation of these cysteines is reported to influence membrane raft localization, although this is controversial as association with Lck may also localize CD4 to membrane rafts (*Crise and Rose, 1992*; *Fragoso et al., 2003*; *Ladygina et al., 2011*). Our analyses suggest that the combination of the CV +C and the GGXXG/S motifs are unique to extant eutherians, co-arose during evolution, and may work together to regulate CD4 membrane localization. Of note, we consider it formally possible that what we are considering here as two distinct motifs may be part of one larger functional motif as the TMD and juxtamembrane regions encompassing GGXXG/S and CV +C are heavily conserved in eutherians.

A poly-basic RHRRR motif in the juxtamembrane region of the ICD was also previously reported to impact human CD4 localization to membrane rafts (*Popik and Alce, 2004*). Our comparative analyses suggest a core HXXR motif (mouse 423–426) with H423 and R426 under purifying selection in the full dataset, and R426 under purifying selection in the mammal-only dataset (*Figure 1A, B* and *Figure 1—figure supplement 1E*).

Further downstream, NMR has shown that the ICD of human CD4 contains a helix-turn struc-ture, the sequence of which is highly conserved in mammalian CD4 molecules (gray shaded region, 429–442, *Figure 1A, B, D*; *Kim et al., 2003*; *Willbold and Rösch, 1996*). A conserved IKRLL motif is embedded within the helix. Its origins likely trace back to the predicted amniote most recent common ancestor (node 1) via the presence of a dileucine repeat. Reptiles and birds diverged away from this motif, while the most recent common ancestor of mammals (node 3) evolved a IXRLL motif that is highly conserved. This region includes several residues under purifying selection within the more limited mammalian dataset (436–440 and 442) or within the full extant CD4 dataset (R430, 434–438 that make up the IKRLL motif, S439, and 440–442 that form the turn) (*Figure 1B* and *Figure 1—figure supplement 1E*). Many of these residues are important for the helix-turn structure, while I434, L437, and L438 are reported to regulate CD4–Lck interactions and endocytosis, suggesting that they are under purifying selection due to their role in a multifunctional hub (*Kim et al., 2003*; *Sleckman et al., 1992*). Given that the helix co-arose with the D3 nonpolar patch that enhances both CD4 and TCR dwell time on pMHCII, as well as pMHCII responses, it is intriguing to speculate that the helix func-tions in part to counterbalance the function of the nonpolar patch (*Glassman et al., 2018*).

Finally, C-terminal to the CQC clasp, mammalian CD4 contains a consensus HRΦQK motif (mouse 448–452 in which Φ represents a large hydrophobic residue; *Figure 1A*). This putative motif is not present in extant fish, reptiles, birds, or even the marsupial CD4 orthologs sequenced to date. Yet, within the mammalian dataset, the codons for H448 and R449 were found to be under purifying selec-tion (*Figure 1B*). The NMR solution structure of the CD4 ICD indicates that this region is unstructured within human CD4 (*Kim et al., 2003*). Given the above analyses, we propose that these residues are likely to be of functional importance.

## Covariation analyses suggest coevolution of motifs in the ECD and ICD

Because some of the motifs considered above co-arose in mammals or eutherians, we explored if residues in these regions showed evidence of covariation. Constraints on protein function can lead to

correlated mutations between residues in a protein that provide further evidence of their functional importance and can highlight networks of functional residues within a protein (*Lockless and Ranganathan, 1999*). We therefore used MISTIC2 to calculate the covariation between residues of CD4 (*Colell et al., 2018*; *Kowarsch et al., 2010*). MISTIC2 quantifies correlations using mutual information as a measure for how much information one random variable provides about another, allowing for detection of covarying relationships between residues that are spatially distant and not just those that are proximal. The exact mechanisms that lead to residue covariation are poorly understood. However, it is widely assumed that the excess of correlated changes in pairs of residues across an evolutionary tree result from molecular coevolution (*Brown and Brown, 2010*; *Capra et al., 2010*; *Dunn et al., 2008*; *Hopf et al., 2015*; *Larson et al., 2000*; *Marks et al., 2011*; *Martin et al., 2005*; *Reynolds et al., 2011*).

By analyzing the full dataset we identified five pairs of covarying residues within the ICD helix-turn region (S432–I434; S432–S439; I434–K441; L437–S439; L437–K442), which may be relevant to the structure of this region, its function, or both (*Figure 1D*). Interestingly, G402 covaries with L438, suggesting covariation between the TMD and ICD. Furthermore, H423 covaries with I434 and S439, and R426 covaries with S439. We also found that P228 and P281 of the nonpolar patch in the D3 domain of the ECD show strong covariation with residues in the ICD helix. Specifically, our data suggest that P228 covaried with S432, I434, S439, and K441, while P281 covaried with S432, L438, and K442. Given that these covarying residues reside in distinct regions that either preclude direct interactions (e.g., ECD, TMD, and ICD), or show no evidence of direct interactions in existing structures (*Kim et al., 2003*), one interpretation of these results is that the covarying residues represent a network of functional motifs that regulate CD4 activity either through the additive impact of their individual functions and/or through allosteric means.

## Functional analysis of motifs

The results above suggest a fitness cost for eutherians if mutations are acquired at residues in the described motifs. Seminal structure–function analyses of CD4 in 58α⁻β⁻ T cell hybridomas established a link between CD4–Lck interactions via the CQC clasp and IL-2 production (*Glaichenhaus et al., 1991*). We therefore performed similar analyses to ask if there is a functional interplay between the transmembrane GGXXG and juxtamembrane CV +C motifs that co-arose in eutherians and may be part of a larger, more continuous functional unit. We also analyzed the IKRLL motif, S432, and S439 residue of the intracellular helix as prior work and our covariation analysis suggested that the intracellular helix may be a multifunctional hub (*Kim et al., 2003*; *Sleckman et al., 1992*).

The goal of our structure function analysis was to change the chemical nature of these CD4 motif and then infer their normal function from the mutant phenotype. We either mutated residues under purifying selection to alanines, reversed charges, changed cysteines to serines, changed serines to alanines to prevent phosphorylation, or changed serines to aspartic acid as a negatively charged phosphomimetic (see *Figure 1C* and *Table 1*, mutant names describe the motif targeted).

We evaluated the impact of these mutations in 58α⁻β⁻ cells transduced to express the 5 c.c7 TCR, which recognizes the moth cytochrome c (MCC 88–103) peptide presented in I-E^k (MCC:I-E^k), and WT or mutant CD4 molecules as per our prior work (*Glassman et al., 2016*; *Glassman et al., 2018*; *Parrish et al., 2016*; *Parrish et al., 2015*). *Supplementary file 1* summarizes the impact of the panel of CD4 mutations studied here, relative to WT, for key biochemical and functional properties.

To study the impact of the mutations on membrane localization, Triton X-100 lysates were sucrose gradient fractionated. Proteins that float on the gradient localize to detergent-resistant membrane (DRM) domains rich in membrane raft components, such as cholesterol and sphingolipids including GM1 (*Pike, 2006*). The remaining proteins localize to detergent-soluble membrane (DSM) domains. We used immunoprecipitation and flow cytometry to quantify the percent of CD4 signal in each fraction, relative to the total, as well as the amount of GM1 or Lck signal in each fraction normalized to the CD4 signal in that fraction (*Fragoso et al., 2003*; *Lee et al., 2021*; *Parrish et al., 2016*). Area under the curve (AUC) was calculated for the DRM (fractions 1–5) and DSM (fractions 6–10) fractions to measure the signal localized to each fraction (*Pike, 2006*; *Sezgin et al., 2017*).

Finally, to study the impact of the mutations on signaling we cocultured the 58α⁻β⁻ cells with I-E^k+ M12 cells and a MCC peptide titration to measure IL-2 production as an endpoint readout of signaling. AUC analysis of IL-2 production allowed us to compare response magnitude between samples while

**Table 1.** Motifs and mutants analyzed in this study.

| Motif location/known function | Mutant names | Mutated motif | Residue mutations |
|---|---|---|---|
| TMD/protein or cholesterol interactions | TMD | GGxxG | G403V, G406L |
| Juxtamembrane/palmitoylation | Palm | CV +C | C418S, C421S |
| TMD + palm/raft localization | TP | GGxxG, CV +C | G403V, G406L, C418S, C421S |
| ICD clasp/interact with Lck, Lat | Clasp | CQC | C444S, C446S |
| TMD + palm + clasp/raft, Lck, Lat interaction | TPC | GGxxG, CV +C, CQC | See TMD + palm + clasp above |
| Total ICD helix | H | Total helix mutation | aa430–442 (to NGPGGNPGGNAGG) |
| Total helix + clasp | HC | Total helix + CQC | aa430–442, C444S, C446S |
| Helix IKRLL only | LL | IKRLL | L437A, L438L |
| Helix serines only | SS | RMSQIKRLLSEKK | S432A, S439A |
| Phosphomimetic helix serines | pSS | RMSQIKRLLSEKK | S432D, S439D |
| Helix IKRLL + serines (does not express) | LL +SS | See LL and SS | L437A, L438L, S432A, S439A |
| Helix IKRLL + phosphomimetic serines | LL + pSS | See LL and pSS | L437A, L438L, S432D, S439D |
| C-terminally truncated CD4 | CD4-T1 | Ends at R422 | R422 is the last residue |
| Extracellular D3 domain nonpolar patch | D3Patch | PXLF | P228E, F231E |
| Extracellular D1 C"-strand (binds pMHCII) | Δbind | GKGVLIR | K68D, V70D, L71S, I72D, R73S |
| IKRLL + D3 nonpolar patch | LL + D3Patch | IKRLL +PXLF | L437A, L438L + P228E, F231E |
| IKRLL + Δbind | LL+ Δbind | See LL + Δbind | L437A, L438L + K68D, V70D, L71S, I72D, R73S |

responses at the lowest peptide dose (41 nM) reported sensitivity. We also analyzed CD4 and TCR endocytosis which are thought to be linked and serve as measures of pMHCII engagement, although the motifs studied here could impact CD4 endocytosis (*Balagopalan et al., 2009*; *Sleckman et al., 1992*). Additionally, we asked if differences in IL-2 production could be linked to differences in proximal pMHCII-specific signaling events by analyzing phosphorylation of key TCR proximal signaling intermediates by flow cytometry (pCD3 ζ , pZap70, and pPlcγ1).

## The GGXXG and CV +C motifs influence CD4 membrane localization and function

First, we asked if the GGXXG and CV +C motifs together influence membrane domain localization and function. We included the CQC clasp motif in this analysis because Lck has myristylation and palmitoylation sites that could influence membrane domain localization of CD4 when associated via the clasp (*Ladygina et al., 2011*). Accordingly, we generated 5 c.c7$^+$ 58α$^-$β$^-$ cells expressing either WT CD4 or the following mutants: TMD, Palm, Clasp, TMD + Palm (TP), TMD + Palm + Clasp (TPC) (*Table 1* and *Figure 2—figure supplement 1*).

To analyze membrane domain localization, we first focused on the percent of CD4 signal in each sucrose gradient fraction, relative to the total, to account for any differences in the amount of CD4 between samples and independent experiments (*Figure 2A* and *Figure 2—figure supplement 2A*). AUC analysis showed that the Palm and Clasp mutants trended lower than WT for DRM localization in our sample size, consistent with prior work (*Fragoso et al., 2003*), while the TP and TPC mutants were significantly reduced. The TP and TPC mutants trended slightly higher in DSMs. These data indicate that the GGXXG plus CV +C motifs together mediate CD4 localization to DRMs.

We next normalized the cholera toxin subunit B (CTxB) signal in each fraction to the CD4 signal in that fraction to assess the amount of GM1 that co-IP'd with CD4 per fraction (*Figure 2B* and *Figure 2—figure supplement 2B*). We did this because membrane rafts are heterogenous in protein and lipid composition, and reasoned that CTxB staining would help us evaluate if our mutations allowed CD4 to remain in membrane rafts, as defined by the DRM fraction, but inhabit different subdomains with different compositions within the DRM fraction (*Pike, 2006*). AUC analysis revealed that the Clasp and

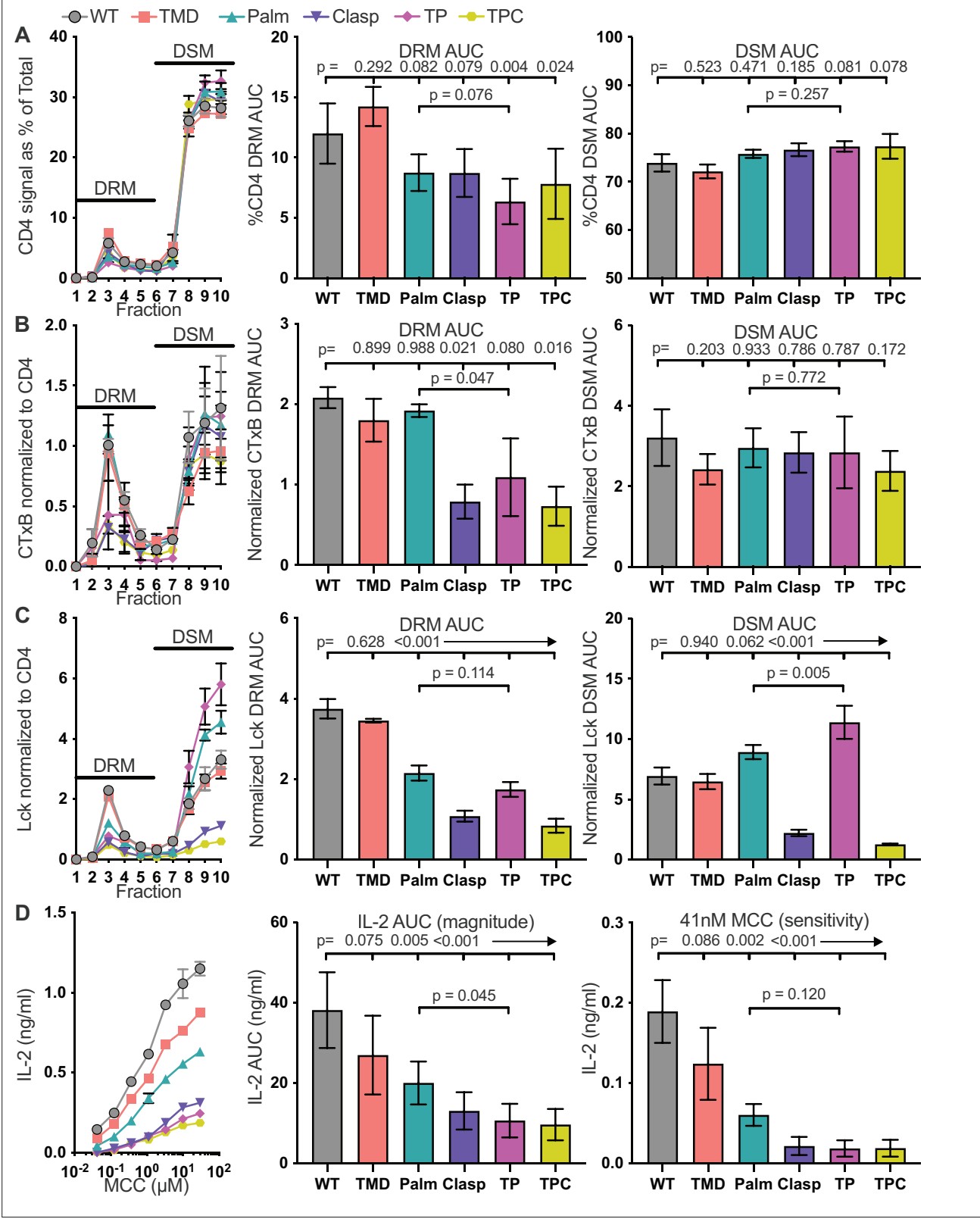

**Figure 2.** The GGXXG + CV+C motifs influence CD4 membrane domain localization and function. (**A**) CD4 signal for each sucrose gradient fraction is shown as a percent of the total CD4 signal detected in all fractions (left). The area under the curve (AUC) is presented for the normalized CD4 signal in the detergent resistant membrane (DRM) fractions (center) and detergent soluble membrane (DSM) fractions (right). (**B**) Cholera toxin subunit B (CTxB) signal is shown for each sucrose fraction normalized to the CD4 signal detected in the corresponding fraction (left). The AUC is shown for the

*Figure 2 continued on next page*

*Figure 2 continued*

normalized CTxB signal in the DRM (center) and DSM (right) fractions. (**C**) Lck signal is shown for each sucrose fraction normalized to the CD4 signal detected in the corresponding fraction (left). The AUC is shown for the normalized Lck signal in the DRM (center) and DSM (right) fractions. (**D**) IL-2 production is shown in response to a titration of MCC peptide (left). AUC analysis for the dose response is shown as a measure of the response magnitude (center). The average response to a low dose (41nM) of peptide is shown as a measure of sensitivity (right).For (A-C) each data point represents the mean ± SEM for the same three independent experiments (biological replicates). For (D), the dose response represents one of three experiments showing the mean ± SEM of triplicate wells (technical replicates). the magnitude and sensitivity data represents the mean ± of three independent experiments (biological replicates). One-way ANOVA with a Dunnet's posttest for comparisons with WT samples, or a Sidak's posttest for comparisons between selected samples, were performed. Key: AUC = Area Under the Curve; DRM = Detergent Resistant Membrane; DSM = Detergent Soluble Membrane; CTxB = Cholera Toxin subunit B.

The online version of this article includes the following figure supplement(s) for figure 2:

**Figure supplement 1.** Surface expression.

**Figure supplement 2.** Raw sucrose gradient values.

**Figure supplement 3.** TCR (left) and CD4 (right) endocytosis after pMHCII engagement is shown for the indicated cell lines after 16 hours coculture with APCs in the presence of 10μM MCC peptide.

**Figure supplement 4.** Total Lck normalized to CD4.

TPC mutants within DRMs had reduced CTxB staining, and the TP mutant had lower CTxB staining than the Palm mutant. There were no noteworthy differences in the DSM fractions. For CD4 molecules within DRMs, the clasp therefore influences CD4 association with GM1-containing membrane subdomains while the GGXXG and CV +C motifs together have a greater influence on subdomain localization than the CV +C motif alone.

We also normalized the Lck signal in each fraction to the CD4 signal detected in that fraction to analyze the amount of Lck that co-IP'd with CD4 per fraction (*Figure 2C* and *Figure 2—figure supplement 2C*). The Palm, Clasp, TP, and TPC signals were all greatly reduced for AUC analysis of the DRM, indicating that both the CV +C and clasp motifs influence association with Lck in DRMs. AUC analysis also showed that only the Clasp and TPC mutants had reduced Lck association in DSMs, whereas the Palm mutant trended higher, and the TP mutant had significantly increased association with Lck relative to the WT. The CQC clasp motif therefore influences CD4–Lck interactions whereas the GGXXG and CV +C motifs together influence the type of membrane domain in which Lck-associated CD4 molecules localize.

To determine how these motifs influence pMHCII responses we measured IL-2 production in response to a titration of MCC peptide. If the frequency of CD4–Lck interactions is the chief determinant for pMHCII responses, then only the Clasp and TPC mutants should reduce IL-2 production as the Palm and TP mutants interacted with Lck in the DSM (*Glaichenhaus et al., 1991*; *Stepanek et al., 2014*). But, if CD4 association with Lck in the DRMs is important, then the Palm and TP mutants would be expected to have reduced IL-2 production. We observed a hierarchy of IL-2 production of WT > TMD > Palm > Clasp ≥ TP ≥ TPC in response to a titration of MCC (*Figure 2D*) that was reflected in AUC analysis. Also, the TP mutant produced less IL-2 than the Palm mutant. The same hierarchy of IL-2 production was observed in response to the lowest dose of MCC tested (41 nM). Of note, only the TPC mutant impacted TCR endocytosis, which is typically a measure of triggered TCRs, while CD4 endocytosis inversely mirrored the normalized CD4–Lck signal in DRMs which either suggests that the CQC motif and GGXXG together with CV +C motif directly impact CD4 endocytosis upon triggering, or that positioning of CD4 in DRMs is important for cointernalization with the TCR (*Figure 2—figure supplement 3*). Overall, the data suggest that the CV +C and GGXXG motifs together enhance pMHCII responses by impacting CD4 membrane domain localization rather than CD4–Lck association. Indeed, we found higher overall CD4–Lck association in the TP cells than the WT (*Figure 2—figure supplement 4* and *Supplementary file 1*), supporting the conclusion that the frequency of CD4–Lck pairs is not the chief determinant of IL-2 responses to agonist pMHCII in this system.

## The intracellular helix interacts with Lck and attenuates pMHCII responses

To study the intracellular helix-turn structure we first replaced residues 430–442 with NGPGGNPGG-NAGG to disrupt the chemical and structural nature of the helix-turn region but maintain its length

(*Table 1*). We also combined this helix (H) mutant with the clasp mutant (HC) to explore how they work together (*Figure 3—figure supplement 1*). Both mutants localized in DRMs and DSMs similar to the WT, both reduced CD4–Lck interactions as expected from prior work, and yet, unexpectedly, both showed a higher magnitude and sensitivity of IL-2 responses to agonist pMHCII than the WT (*Figure 3* and *Figure 3—figure supplements 2 and 3*; *Kim et al., 2003*; *Sleckman et al., 1992*). We also observed more TCR endocytosis for the H mutant than the WT or HC mutant, indicating that the increased IL-2 output by the H mutant might reflect more triggered TCRs over the course of 16 hr. Finally, because CD4 can increase TCR dwell time on pMHCII, the failure of the H and HC mutant CD4 molecules to endocytose over the course of 16 hr of 58α⁻β⁻ cell co-culture with antigen-presenting cells (APCs) could result in more sustained signaling in that time period and partially explain the increased IL-2 (*Figure 3—figure supplement 4*; *Glassman et al., 2018*; *Sleckman et al., 1992*).

## The IKRLL motif and flanking serines regulate pMHCII responses

To determine if mutating specific residues within the helix would mimic the loss of the helix (H mutant) we mutated the dileucine repeat (L437A + L438A = LL mutant) within the IKRLL motif of the ICD helix (*Table 1* and *Figure 1C*). I/LXXLL motifs and dileucine repeats are known protein interaction mediators, and structural data indicate that they contribute to CD4 ICD helix interaction with Lck and AP-2 (*Kelly et al., 2008*; *Kim et al., 2003*). We also mutated the intracellular helix serines because (de) phosphorylation at one or both residues may regulate function (*Sleckman et al., 1992*). Our SS mutant (S432A.S439A) was designed to prevent phosphorylation or any interactions involving the hydroxyl groups, while the negatively charged pSS mutant (S432D.S439D) was used to mimic phosphorylation at these residues. We also combined mutations (LL + SS and LL + pSS) to infer how these residues may work together within the helix given their covariation over evolutionary time (*Figure 1D*). Of note, the LL + SS mutant did not express on the cell surface and thus was not analyzed further. The SS mutant had lower surface expression than the WT and the LL + pSS expression was slightly reduced (*Figure 4—figure supplement 1*).

For these lines, the SS and LL + pSS showed a decreased percent of CD4 localized in DRMs compared with the WT, yet none of the mutations significantly changed the percent of CD4 signal localized to DSMs. Furthermore, none of the mutations impacted CD4-associated CTxB signal in DRMs, although the LL + pSS trended lower, and only the pSS and LL + pSS mutants reduced the amount of CTxB signal associated with CD4 in DSMs (*Figure 4A, B* and *Figure 4—figure supplement 2A and B*). These data suggest that the IKRLL motif alone does not influence membrane domain localization, but that the hydroxyl group on the serine residues and a negative charge at these positions can influence membrane domain localization.

Regarding CD4–Lck association, we found that the Lck signal associated with the LL mutant was reduced in the DRM fraction (*Figure 4C* and *Figure 4—figure supplement 2C*). Interestingly, CD4–Lck association trended lower for the pSS mutant than the WT and was lower than the SS mutant in DRMs. Within DSMs, the LL trended lower than the WT, the SS mutant was greatly increased over the WT, and the pSS mutant was equivalent to the WT. These data extend prior work indicating that the IKRLL motif of the ICD helix mediates CD4–Lck interactions while S432 and/or S439, which do not contact Lck directly in the NMR structure, play a role in regulating CD4–Lck association at the helix (*Kim et al., 2003*; *Sleckman et al., 1992*).

For IL-2 we found that both the responses magnitude and sensitivity to agonist pMHCII were greatly increased for the LL mutant compared with the WT even though the total amount of CD4-associated Lck was lower (*Figure 4D* and *Figure 4—figure supplement 3*). In contrast, the SS line had higher total CD4-associated Lck than the WT yet the magnitude and sensitivity of IL-2 responses were lower. These data further support the idea that CD4–Lck interactions are not the chief determinant of IL-2 responses to agonist pMHCII in this system. Also of note, both the pSS and LL + pSS mutants had equivalent IL-2 responses to the WT (*Figure 4D*). Together, the data suggest that the IKRLL motif functions to inhibit the magnitude and sensitivity of pMHCII responses and that phosphorylation of the flanking serines regulate this activity.

We also evaluated if differences in IL-2 output by these mutant cells could be correlated with changes in TCR or CD4 surface levels during stimulation with APCs. While there was a trend toward more TCR internalization with the LL mutant, and less with the SS mutant, we could not attribute differences in IL-2 production to TCR triggering (*Figure 4—figure supplement 4A*). The LL, SS, pSS,

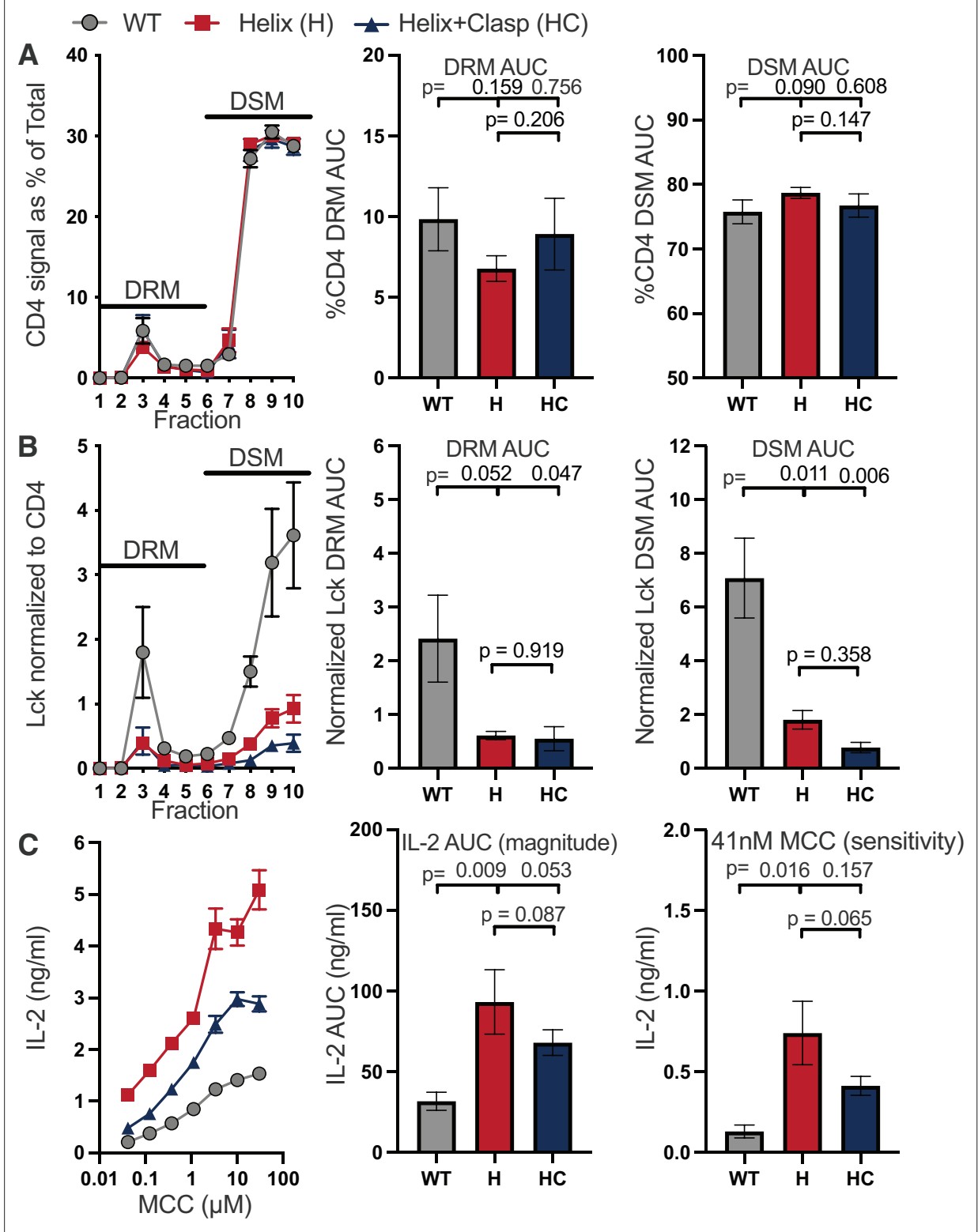

**Figure 3.** The intracellular helix attenuates response magnitude and sensitivity. (**A**) CD4 signal for each sucrose gradient fraction is shown as a percent of the total CD4 signal detected in all fractions (left). The area under the curve (AUC) is presented for the normalized CD4 signal in the detergent resistant membrane (DRM) fractions (center) and detergent soluble membrane (DSM) fractions (right). (**B**) Lck signal is shown for each sucrose fraction normalized to the CD4 signal detected in the corresponding fraction (left). The AUC is shown for the normalized Lck signal in the DRM (center) and DSM (right) fractions. (**C**) IL-2 production is shown in response to a titration of MCC peptide (left). AUC analysis for the dose response is shown as a measure

*Figure 3 continued on next page*

*Figure 3 continued*

of the response magnitude (center). The average response to a low dose (41nM) of peptide is shown as a measure of sensitivity (right). For (A–C) The data are presented as in *Figure 2*. Key: AUC = Area Under the Curve; DRM = Detergent Resistant Membrane; DSM = Detergent Soluble Membrane.

The online version of this article includes the following figure supplement(s) for figure 3:

**Figure supplement 1.** Surface expression.

**Figure supplement 2.** Raw sucrose gradient values.

**Figure supplement 3.** Total Lck Normalized to CD4.

**Figure supplement 4.** TCR and CD4 endocytosis.

and LL + pSS mutants all failed to endocytose CD4 to the same extent as the WT, with no internalization being observed for the LL and LL + pSS mutants after 16 hr. The lack of a correlation between IL-2 responses and TCR or CD4 endocytosis for these helix mutants pointed to the CD4 ICDs as being responsible for directing different IL-2 outputs between these mutants. To test this further we generated 58α⁻β⁻ cells expressing either CD4 WT, a C-terminally truncated mutant that ends at R422 (CD4-T1), or the LL mutant (*Figure 1* and *Table 1*). We reasoned that CD4-T1, which was previously reported to relieve CD4−Lck interactions and diminish IL-2, should not be endocytosed during co-culture with APCs because it lacks the ICD helix (*Glaichenhaus et al., 1991*) therefore, if the CD4-T1 mutant fails to endocytose upon stimulation yet directs reduced IL-2 responses relative to the LL mutant, then the IL-2 responses directed by the mutants in *Figure 4D* can be directly attributed to the ICD and not CD4 levels. We found that CD4-T1 expressed at similar levels to LL, failed to internalize upon stimulation, and directed reduced IL-2 responses relative to the LL mutant (*Figure 4—figure supplement 5A–C*). These data further support the conclusion that differences in IL-2 production in *Figure 4D* are directed by differences in the CD4 ICDs more than cell surface levels.

Finally, we performed ELISpot to ask if the difference in IL-2 production between the WT and LL mutant cells was due to an increased frequency of responders making IL-2. We observed more responders for the LL mutant cells than the WT for two independently generated lines (*Figure 4—figure supplement 6*). The average spot intensity was also higher for the LL mutant in one of the two lines tested, and trended higher for the other, which suggest each cell made more IL-2 within the assay period. The simplest interpretation of these data, when considered with the increased sensitivity and response magnitude measured by ELISA, is that the LL mutation lowers the signaling threshold that must be overcome for IL-2 production.

## Evidence for counterbalancing functions between CD4 motifs

The data in *Figure 4D* corroborated the functional link between S432 and/or S439 and the IKRLL motif predicted by our covariation analysis (*Figure 1C*), which also predicted a link between the intracellular helix and the ectodomain D3 nonpolar patch that arose in the predicted mammalian most recent common ancestor (*Figure 1—figure supplement 1E*). We hypothesized that the advantage gained from the ability of CD4 to stabilize TCR–CD3–pMHCII interactions and increase signal strength necessitated the coevolution of elements with the ability to regulate the enhanced signaling capacity. Alternatively, the inhibitory function of the helix allowed for the evolution of the nonpolar patch. Regardless, these data suggest a functional counterbalancing action between both motifs. Accordingly, we combined an ELXE mutant (P228E + F231E) of the PLXF motif in the D3 domain, which reduces 58α⁻β− IL-2 responses (*Glassman et al., 2018*), with the LL mutation (*Figure 5—figure supplement 1*) to ask if the ICD helix regulates the increased signaling afforded by the nonpolar patch. As a control we combined a GKGVLIR to GDGDSDS mutant in the D1 domain (CD4^Δbind, *Figure 1C* and *Table 1*), which kills CD4 binding to pMHCII (*Glassman et al., 2016*; *Parrish et al., 2015*), to confirm that the LL mutant phenotype is dependent on CD4–pMHCII interactions. We found that the LL + ELXE double mutant drove similar IL-2 response magnitude and sensitivity to agonist pMHCII as the WT, while the LL + Δbind double mutant completely impaired responses (*Figure 5A, B*). Therefore, the intracellular helix and IKRLL motif therein do not regulate pMHCII-independent activity of CD4 in our system; rather, they counterbalance the formation of a stable TCR–CD3–pMHCII–CD4 assembly mediated by the ectodomain nonpolar patch.

Our covariation analysis also suggested a functional link between the GGXXG motif in the TMD with the intracellular helix. Because the TP mutant severely reduced the magnitude and sensitivity of

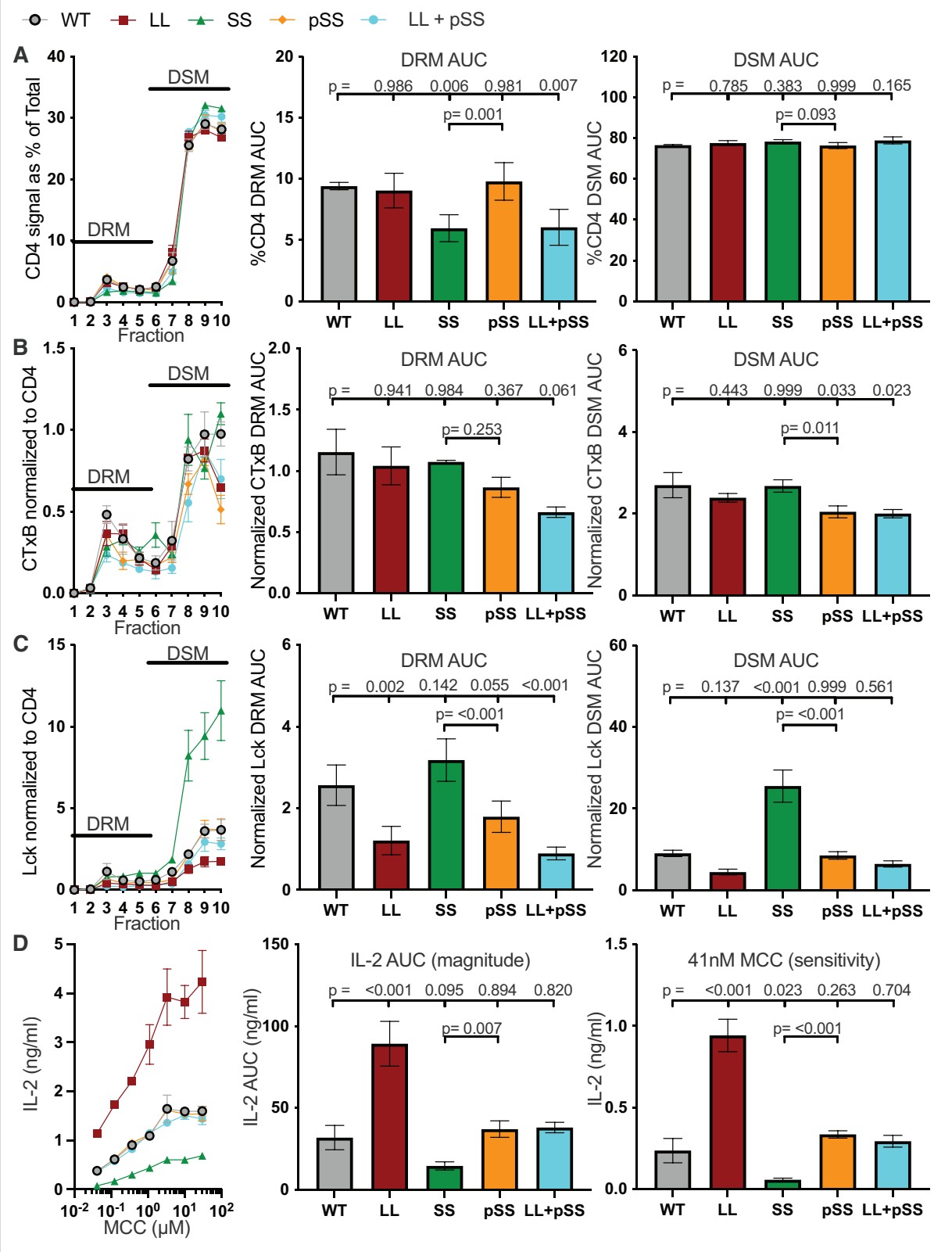

**Figure 4.** The IKRLL motif mediates the inhibitory function of the helix. (**A**) CD4 signal normalized as a percent of the total is shown for each sucrose gradient fraction (left) along with the AUC analysis for the DRM (center) and DSM (right) fractions. (**B**) Cholera toxin subunit B (CTxB) signal normalized to CD4 signal detected is shown for each sucrose fraction (left) along with the AUC analysis for the DRM (center) and DSM fractions (right). (**C**) Lck signal normalized to CD4 signal is shown for each sucrose fractions (left) along with the AUC analysis for the DRM (center) and DSM (right) fractions.

*Figure 4 continued on next page*

*Figure 4 continued*

(**D**) IL-2 dose response to MCC peptide (left). AUC analysis as a measure of the response magnitude (center), and the average response to a low dose (41nM) of MCC as a measure of sensitivity (right) are shown. For (A–D), the data are presented as in *Figure 2*. Key: AUC = Area Under the Curve; DRM = Detergent Resistant Membrane; DSM = Detergent Soluble Membrane; CTxB = Cholera Toxin subunit B.

The online version of this article includes the following figure supplement(s) for figure 4:

**Figure supplement 1.** Surface expression.

**Figure supplement 2.** Raw sucrose gradient values.

**Figure supplement 3.** Total Lck normalized to CD4.

**Figure supplement 4.** TCR and CD4 endocytosis.

**Figure supplement 5.** Comparison of CD4 T1 and LL mutants.

**Figure supplement 6.** ELISPOT analysis.

IL-2, we combined it with the LL mutation (LL + TP) to ask if these mutations counterbalance each other. We found that cells expressing the double mutant had similar IL-2 responses to the WT, which were lower than the LL cells only, indicating that motifs within the TMD and ectodomain can exert counterbalancing activities on pMHCII responses (*Figure 5A, B*). As the GGXXG and CV +C motifs co-arose in eutherians after the intracellular helix and nonpolar patch, these data points to additional pressure to evolve motifs with the capacity to regulate CD4 function by regulating its contribution to pMHCII-specific signaling.

## Distinct CD4 motifs differentially impact TCR–CD3 signal transduction

Because IL-2 production is an endpoint readout for signaling, we also asked if the IL-2 phenotypes of the Clasp, TP, and LL mutants could be attributed to defects in proximal signaling events. Accordingly, we analyzed phosphorylation of CD3 ζ and Zap70, both Lck substrates, as well as Plcγ1 which is phosphorylated by ITK after it is activated by Lck (*Figure 2—figure supplement 4* and *Figure 4— figure supplement 3*; *Courtney et al., 2018*; *Gaud et al., 2018*). If the abundance of CD4–Lck pairs is directly related to the magnitude of these signaling steps, then the Clasp and the LL mutants should have lower pCD3 ζ , pZap70, and pPlcγ1 levels compared to the WT because the mutations reduced total CD4–Lck abundance by ~31% and ~49% of WT levels, respectively, while the TP mutant should have increased levels of pCD3 ζ , pZap70, and pPlcγ1 because this mutant increased total CD4–Lck abundance to 123% of the WT (*Supplementary file 1*; *Glaichenhaus et al., 1991*; *Rudd, 2021*; *Stepanek et al., 2014*). Alternatively, if CD4 sequesters Lck away from TCR–CD3 until pMHCII engagement to prevent signal initiation by free Lck, and free Lck is more active than CD4-associated Lck, then the Clasp and LL mutants should have equivalent or higher pCD3 ζ , pZap70, and pPlcγ1 levels than the WT due to free Lck while the TP mutant should have either equivalent or reduced levels due to higher CD4–Lck interactions and sequestration (*Van Laethem et al., 2007*; *Wei et al., 2020*).

To test these predictions, we analyzed pCD3 ζ , pZap70, and pPlcγ1 levels by flow cytometry for TCR⁺ CD4⁺ 58α⁻β⁻ cells coupled to APCs expressing either the null peptide hemoglobin 64–76 (Hb) tethered to I-Eᵏ (Hb:I-Eᵏ) or the agonist MCC peptide tethered to I-Eᵏ (MCC:I-Eᵏ) (*Figure 6—figure supplement 1A–C*). This approach allowed us to evaluate the impact of the CD4 motifs studied here on proximal signaling initiated by engagement of cognate ligand, which cannot be achieved with conventional anti-CD3 antibody crosslinking approaches, while the high ligand density of tethered MCC:I-Eᵏ allowed for rapid synchronous engagement of TCRs to monitor proximal signaling events similar to conventional antibody-induced signaling (i.e., the TCRs did not have to find agonist peptide among irrelevant pMHCII on peptide-pulsed APCs). One caveat to this approach is that the high density of MCC:I-Eᵏ might mask CD4 contributions that we and others have reported to be more apparent for responses to low densities of agonist pMHCII (*Glassman et al., 2018*; *Irvine et al., 2002*). However, this concern is somewhat mitigated by prior work showing that IL-2 production with this experimental setup is CD4 dependent (*Parrish et al., 2016*). Moreover, we found that TCR⁺ CD4⁺ 58α⁻β⁻ cells bearing the Clasp and TP mutants made less IL-2 than those bearing the WT in response to APCs expressing tethered MCC:I-Eᵏ, while cells bearing the LL mutant made more IL-2 than the WT (*Figure 6—figure supplement 2*). These data suggest that the Clasp, TP, and LL mutations similarly impact the signaling pathways that lead to IL-2 production, be it in response to low or high densities of agonist pMHCII.

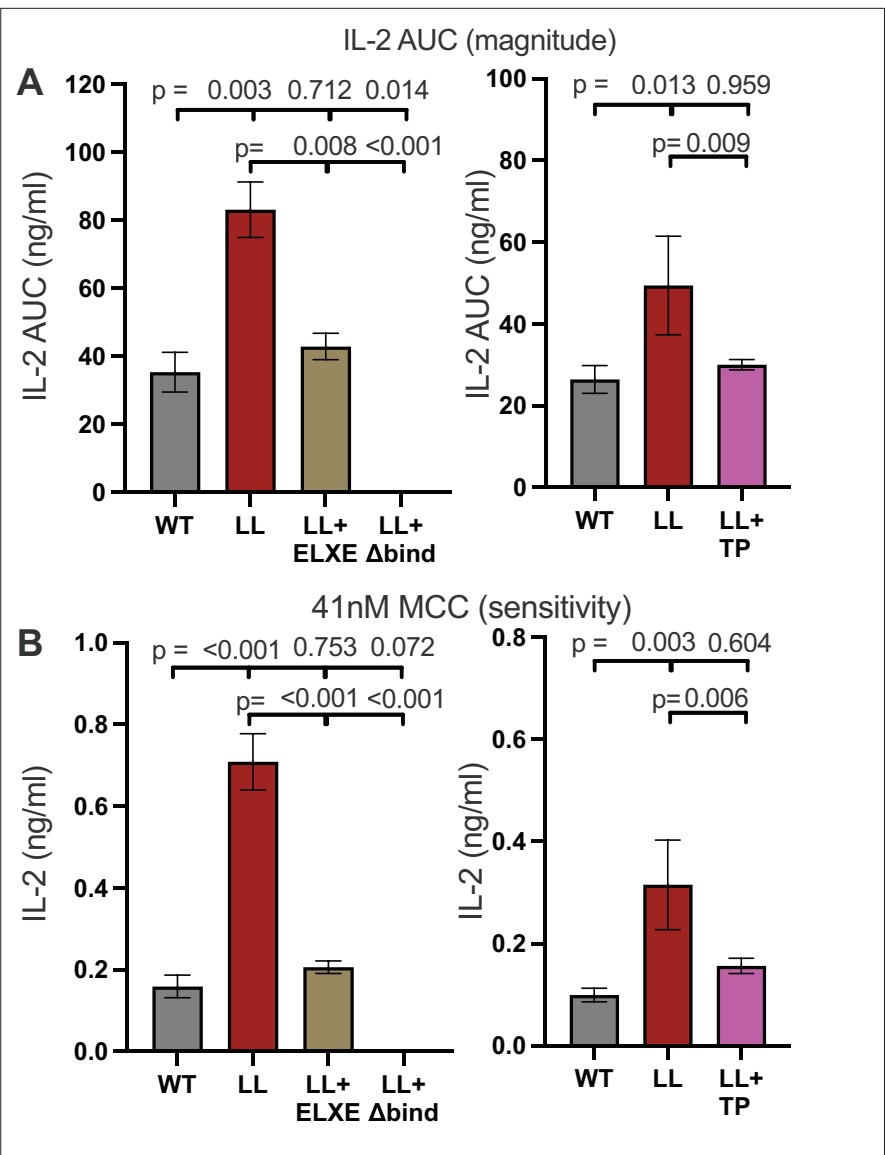

**Figure 5.** Coevolving motifs in the extracellular and intracellular domains functionally counterbalance each other. (**A**) AUC analysis of IL-2 dose response to MCC peptide are shown as a measure of the response magnitude for the indicated samples. (**B**) The average IL-2 response to a low dose (41nM) of MCC is shown as a measure of sensitivity for the indicated samples. For (A and B) the magnitude and sensitivity data represent the mean ± SEM of three independent experiments (biological replicates) for which triplicate measurements were performed (technical replicates). One-way ANOVA was performed with a Dunnett's posttest for comparisons with WT samples, and a Sidak's posttest for comparisons between selected samples. Individual graphs indicate experiments that were performed with cell lines generated at the same time. Key: AUC = Area Under the Curve.

The online version of this article includes the following figure supplement(s) for figure 5:

**Figure supplement 1.** IL-2 production and surface expression.

For the paired WT and Clasp, WT and TP, and WT and LL cell lines in *Figure 6* marked by solid symbols we performed three independent experiments, collecting 10,000 coupled cells per experiment, and concatenated the flow cytometry data prior to further analysis. For those paired WT and LL mutant cells marked by open symbols, we performed the experiment once each. For data processing, we subtracted the MCC:I-E$^k$ phospho-protein intensity from the Hb:I-E$^k$ intensity to determine the percent of coupled cells that responded to agonist pMHCII. We then compared the mean fluorescence intensity (MFI) of the WT responders to the mutants to evaluate differences in the intensity of

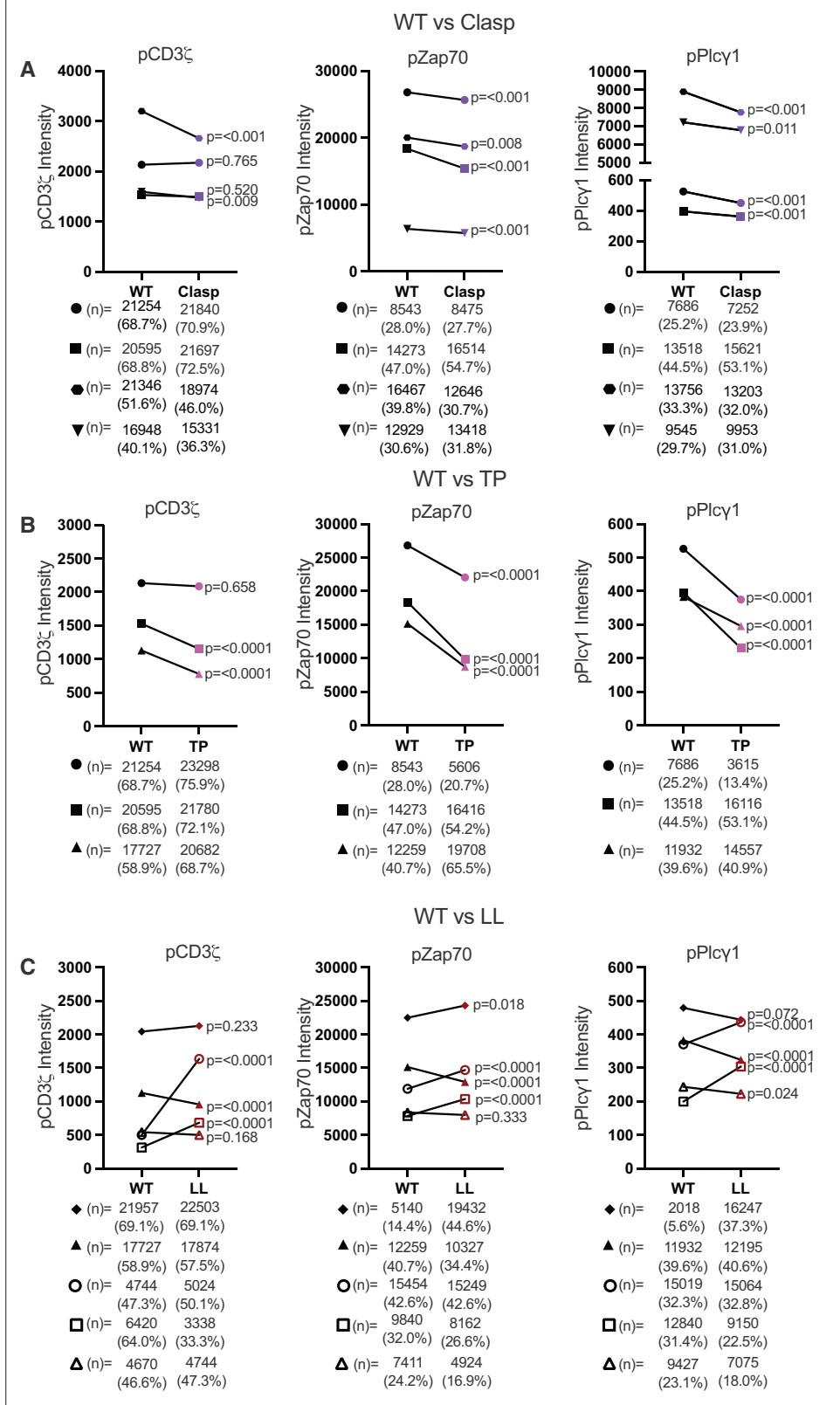

**Figure 6.** The CQC, GGXXG + CV+C, and IKRLL motifs differentially impact proximal TCR-CD3 signaling. (**A**) Phosphorylation intensity for CD3 ζ (left), Zap70 (center), and Plcγ1 (right) are shown for paired (connecting line) WT and Clasp mutant cell lines. Four independently generated cell lines were tested. (**B**) Phosphorylation intensity for CD3 ζ (left), Zap70 (center), and Plcγ1 (right) are shown for paired (connecting line) WT and TP mutant cell

*Figure 6 continued on next page*

*Figure 6 continued*

lines. Three independently generated cell lines were tested. (**C**) Phosphorylation of CD3 ζ (left), Zap70 (center), and Plcγ1 (right) are shown for paired (connecting line) WT and LL mutant cell lines. Five independently generated cell lines were tested. For (A–C), filled symbols represents the mean ± SEM of concatenated data for coupled cells from three independent experiments. 10,000 coupled cells were collected per experiment (technical replicates), resulting in the concatenation of 30,000 coupled cells total from the 3 independent biological replicates. For (C), the open symbols represent the mean and ± SEM for one single experiment (10,000 coupled cells analyzed). One-way ANOVA was performed with a Dunnett's posttest when the experiments involved multiple comparisons. Student's t-test were performed for when only WT and mutant pairs were analyzed in an experiment. The derived p values for each independent cell line comparing the mutant CD4 to its paired WT is shown. Next to each symbol the number of cells determined to have responded to stimuli are shown with the percentage of responding cells.

The online version of this article includes the following figure supplement(s) for figure 6:

**Figure supplement 1.** Proximal signaling analysis workflow.

**Figure supplement 2.** IL-2 production to agonist tethered pMHCII+ APCs.

**Figure supplement 3.** Coupling frequency.

**Figure supplement 4.** Proximal signaling and IL-2 production in response to weak agonist pMHCII for WT and Clasp mutant cell lines.

---

the response. We also compared the frequency of couples, which was unaffected by the mutations (*Figure 6—figure supplement 3*).

For the Clasp mutation, we found no difference in the pCD3 ζ MFI compared to the WT for two independently generated cell lines, for a third line the pCD3 ζ MFI was reduced to ~83% of WT, and for a fourth line we saw a small but statistically significant reduction to ~92% of WT (*Figure 6A*). There was no obvious impact on the percent of responders. Interestingly, pZap70 and pPlcγ1 MFI were significantly lower for all four Clasp lines compared to their respective WTs, despite no clear difference in percent responders. The most parsimonious interpretation of these data is that, as tested, reducing CD4–Lck interactions by mutating the CQC clasp does not prevent, or consistently reduce, pCD3 ζ phosphorylation but does reduce the phosphorylation of other Lck substrates. The Clasp mutation therefore did not impact pCD3 ζ levels in response to agonist pMHCII as predicted by the TCR signaling paradigm.

For the TP mutant, pCD3 ζ MFI levels were significantly lower than WT for two of three independently generated cell lines tested in response to the agonist pMHCII, MCC:I-E$^k$, while all three TP lines had slightly higher percent responders for pCD3 ζ than their paired WTs (*Figure 6B*). For pZap70 and pPlcγ1, all sets of lines showed reduced MFI for the TP mutant compared with the WT without impacting the percent responders. The simplest interpretation of these data is that regulation of membrane domain localization of CD4–Lck pairs by the GGXXG and CV +C motifs can influence pCD3 ζ, pZap70, and pPlcγ1 levels in our system. We therefore take these data as evidence that when the CQC clasp is intact, CD4–Lck pairs must be localized in the appropriate membrane compartment for efficient phosphorylation of CD3 ζ.

For the LL mutant, we observed no consistent difference in pCD3 ζ, pZap70, or PLCγ1 MFI or percent responders that would allow us to attribute the increase in IL-2 for the LL mutant to proximal signaling differences in response to the agonist pMHCII, MCC:I-E$^k$, at least not within the 2 min at which these events were evaluated (*Figure 6C*). These data suggest the interactions regulated by the intracellular helix, including CD4–Lck interactions, do not consistently impact early signaling events as measured here.

Finally, because the Clasp mutation did not reduce pCD3 ζ levels in response to agonist pMHCII as predicted by the dominant paradigm, we asked if we would see differences in proximal signaling in response to weak stimulus. Specifically, we and others have shown that IL-2 production by 5 c.*c*7 CD4$^+$ T cells in response to low doses of agonist MCC peptide is CD4 dependent, as are 5 c.*c*7 CD4$^+$ T cell and 5 c.c.7$^+$ CD4$^+$ 58α$^-$β$^-$ cell IL-2 responses to the low affinity, weak agonist altered peptide ligand T102S (*Glassman et al., 2018*; *Irvine et al., 2002*; *Parrish et al., 2016*). We therefore compared proximal signaling of our Clasp mutant and WT cells using APCs expressing a tethered T102S:I-E$^k$ (*Figure 6—figure supplement 4A*). Two of the lines showed no difference in pCD3 ζ MFI levels between the Clasp mutant and paired WT, another had a pCD3 ζ MFI for the Clasp mutant that was ~117% of the WT, and the last line had pCD3 ζ MFI for the Clasp mutant that was ~80% of

WT. There were also no clear differences in the percent of responders. Further, we saw no consistent difference in pZap70 MFI or percent responders between the WT and Clasp mutants in response to the weak stimuli. Finally, for pPlcγ1 MFI, three of the four cell lines had reduced MFI for the Clasp mutants compared with their WT controls. However, for the fourth line the Clasp mutant had higher pPlcγ1 levels than the WT control. The percent responders also trended higher in three of the four Clasp lines. Importantly, the IL-2 responses to APCs expressing the tethered T102S:I-E$^k$ were lower for the Clasp mutants in all four sets of lines relative to their paired WT controls (*Figure 6—figure supplement 4A*). These data are not consistent with predictions of the TCR signaling paradigm.

## Discussion

The goal of this study was to gain novel insights into how ~435 million years of natural selection shaped eutherian CD4 function. We therefore used evolutionary and covariation analyses to guide structure–function studies of mouse CD4 in 58α$^-$β$^-$ T cell hybridomas. We note that while these cells may lack some elements of signal transduction found in real T cells, their IL-2 responses to pMHCII are CD4 dependent so relevant signaling pathways are intact (*Glassman et al., 2018*). Furthermore, protein–protein interactions between the CD4 motifs studied here with their interacting partners should be the same at a biochemical level as in real T cells provided the interacting partners are expressed. Indeed, the link between the CQC clasp motif, CD4–Lck interactions, and IL-2 production were established in seminal work using 58α$^-$β$^-$ cells (*Glaichenhaus et al., 1991*). Finally, because expression levels of CD4, Lck, adaptor proteins, or other molecules that might interact with CD4 vary between phenotypically different T cell populations (http://immpres.co.uk/), the motifs studied here may affect different outcomes in different T cell subsets. Our data are therefore likely to reflect general principles concerning the function of the motifs we identified, even if they may not reflect exactly how the motifs uniquely influence thymocyte, naive, Th, Treg, or Tm cell behavior.

A central tenet of the TCR signaling paradigm, and related models, is that CD4–Lck interactions via the CQC clasp allow CD4 to recruit Lck to phosphorylate the CD3 ITAM upon pMHCII engagement (*Glaichenhaus et al., 1991*; *Rudd, 2021*; *Stepanek et al., 2014*). However, when we tested this model directly with our Clasp mutant, we did not observe consistent reductions in CD3ζ phosphorylation as expected in response to agonist or weak agonist pMHCII. We did observe reduced Zap70 and Plcγ1 phosphorylation in response to agonist pMHCII, which helps explain the reduced IL-2 responses reported here and elsewhere for CQC clasp mutants (*Glaichenhaus et al., 1991*). We interpret these data as evidence that, in our system, Lck can efficiently phosphorylate CD3ζ ITAMs, but not other substrates (e.g., Zap70), even if CD4–Lck abundance is reduced by mutating the CQC clasp. We cannot, however, rule out that mutating the CQC clasp would impact CD3ζ phosphorylation in response to low densities of agonist pMHCII where CD4 is known to be critical for downstream signaling responses such as calcium mobilization or IL-2 production (*Glassman et al., 2018*; *Irvine et al., 2002*); however, on this point it is worth noting that our previous work points to an essential role for the CD4 ectodomain in mediating sensitivity to weak agonist pMHCII, as well as low doses of agonist pMHCII (*Glassman et al., 2018*). Overall, a key takeaway from the results with our Clasp mutant is that focusing solely on CD4–Lck interactions via the CQC clasp fails to convey a full understanding of the functional significance of CD4–Lck pairs.

Indeed, the Clasp mutant is better understood when considered with our TP mutant, which was nearly identical to the Clasp mutant regarding IL-2 responses even though it had ~123% more total CD4–Lck pairs than WT CD4. Enrichment of CD4–Lck pairs in the DSMs of TP mutants corresponded with lower CD3ζ, Zap70, and Plcγ1 phosphorylation, providing evidence that the Clasp and TP mutant IL-2 phenotypes are similar for different reasons. We favor the following interpretation: (1) Lck freed by our Clasp mutant can phosphorylate CD3ζ ITAMs but cannot as efficiently phosphorylate other substrates; (2) CD4–Lck pairs sequestered to the wrong membrane compartment by our CD4 TP mutation does not efficiently phosphorylate CD3ζ ITAMs and other substrates. Taken together, we think the simplest interpretation of these data is that CD4 association with Lck regulates the phosphorylation of Lck targets, including ITAMs, because membrane domain localization of CD4–Lck pairs is regulated. This interpretation supports a variant of the TCR signaling paradigm in which CD4 sequesters Lck away from TCR–CD3 to prevent spurious signaling until reciprocal engagement of pMHCII by TCR–CD3 and CD4 allows Lck recruitment to the CD3 ITAMs (*Glassman et al., 2018*; *Van Laethem et al., 2007*). Varying CD4 palmitoylation would then allow for tuning of CD4 function.

Comparing the Clasp mutant with our intracellular helix mutants provides additional insights into the relationship between CD4–Lck interactions and CD4 function. First, although our LL mutation reduced CD4–Lck interactions to half of WT levels, we again did not observe clear evidence of reduced CD3 ζ phosphorylation. Second, if we consider that only a small fraction of WT CD4 molecules are naturally paired with Lck (~6% in 58α⁻β⁻ cells) then we can intuit that the intracellular helices of those CD4 molecules that are not paired with Lck are free to mediate other function (*Parrish et al., 2016*; *Stepanek et al., 2014*). If we further consider that our Clasp mutant has an intact IKRLL motif within the intracellular helix that is generally not occupied by Lck, and is thus free to mediate those other functions, then we can infer from the increased IL-2 production driven by our H, HC, and LL mutants (which have ~25%, ~14%, and ~49% of WT levels of CD4–Lck association, respectively, but disrupted IKRLL motifs) that a key function of the IKRLL motif is to prevent CD4 molecules that are not paired with Lck from driving pMHCII-specific signaling. Indeed, comparing our Clasp and LL mutant signaling phenotypes implies that one potential function of the IKRLL motif is to inhibit free Lck from phosphorylating substrates other than ITAMs (e.g., Zap70). This would explain the conundrum of why our H, HC, and LL mutants relieve CD4–Lck interactions and yet increase IL-2 responses. Together, we consider the simplest interpretation of our data to be that the IKRLL motif of the intracellular helix is under purifying selection because it mediates CD4–Lck interactions, CD4 endocytosis, and performs a previously unreported inhibitory function. Our SS, pSS, and LL + pSS mutants all suggest that the phosphorylation states of the helix serines regulates these functions. We take these data as evidence that the multifunctional intracellular helix is a key regulator of pMHCII responses.

This conclusion is further supported by our finding of evolutionary covariation between proximal as well as distant residues and motifs. One example is the covariation of proximal residues within the intracellular helix, such as the serines and IKRLL motif, that regulate CD4–Lck interactions and CD4 inhibitory activity. For more distant residues, the GGXXG and CV +C co-arose in the predicted most recent common ancestor of eutherians, have been maintained under purifying selection, and together regulate CD4–Lck membrane domain localization. They also counterbalance the inhibitory activity of the ICD helix, suggesting a functional link that is supported by residue covariation between the GGXXG motif of the TMD and ICD helix. Moreover, our analyses provide evidence that interactions mediated by the ectodomain nonpolar patch, which enhances signaling by stabilizing TCR–CD3–pMHCII–CD4 assemblies on the outside of the cell, are coupled with the intracellular helix that regulates signaling (*Glassman et al., 2018*). Key residues in these motifs covaried over evolutionary time, the motifs co-arose in the predicted most recent common ancestors of mammals, and they have been conserved in marsupials and eutherians under purifying selection. Additionally, our data show that the intracellular helix and nonpolar patch counterbalance each other with regard to IL-2 production. The broader conclusion from these results is that the emergence of the CD4 intracellular helix, with its ability to counterbalance the signal-enhancing activity of other motifs, was a key moment in the regulation of mammalian CD4 function.

Understanding how these motifs work individually to influence eutherian CD4 function and CD4⁺ T cell biology in fine molecular detail represents fertile ground for future directions. Because GXXXG and GG motifs can interact with cholesterol, and palmitate can interact with protein-bound cholesterol (PDB 4IB4), coupling of rapidly reversible palmitoylation of the CV +C motif with a cholesterol-bound state of the GGXXG motif may allow finer regulation of CD4 membrane domain localization and function than can be achieved with a palmitoylation motif alone (*Fessler, 2016*; *Song et al., 2014*; *Teese and Langosch, 2015*; *Wacker et al., 2013*). Structural analysis will help test this hypothesis. For the intracellular helix, the IKRLL motif clearly has multiple interacting partners that dictate its activity. Some are known, such as Lck and AP2 (*Kelly et al., 2008*; *Kim et al., 2003*; *Sleckman et al., 1992*); yet our data lead us to hypothesize that the inhibitory activity of the helix is the result of one or more additional partners. The data here and elsewhere suggest that the preference of partners is likely to be regulated both by their relative abundance, membrane domain localization, and the phosphorylation state of the helix serines. In addition, given that T443 just outside the helix has covaried over evolution with L438 in the IKRLL motif as well as Q445 of the CQC clasp motif, it is reasonable to speculate that the phosphorylation state of this residue might further regulate interactions between the helix and/or clasp and their binding partners. Moreover, given that the CQC clasp links CD4 to Lck by coordinating a zinc, it is reasonable to consider if and how changes in zinc concentration regulate CD4–Lck interacts for their own function. Importantly, zinc-regulated changes in Lck association

would impact occupation of the IKRLL motif by Lck, which would impact the ability of the intracellular helix to interact with other partners to exert other activity. In sum, CD4 function is likely to be regulated by the switch-like activity of multiple motifs in its ICD.

Future studies aimed at understanding how these motifs work together will allow us to better understand why some of them have covaried over evolutionary time. Two plausible, nonmutually exclusive modes can be envisioned for how individual motifs may work together within the network of counterbalancing activities described here. First, given the switch-like activity of the intracellular helix serines, palmitoylation sites in the CV +C motif, and zinc coordination of the CQC clasp, CD4 activity might be finetuned by the sum of the switch states of each motif in the way that the sum of digital states determines a computational output. Second, allosteric effects may also be at play. For example, extracellular interaction mediated by the D3 nonpolar patch might induce an allosteric change in the transmembrane GGXXG motif or intracellular helix that might impact binding partner preferences. These possibilities, which are not mutually exclusive, suggest just how complex the regulation of CD4 may be and give insights into the work that will be required to describe exactly how CD4 functions.

In closing, our multidisciplinary results highlight a network of function-regulating motifs within eutherian CD4 that would not otherwise be obvious. In so doing we extend a theme of counterposing activities that regulate eutherian pMHCII responses at the population level (e.g., helper and regulatory CD4$^+$ T cells) and cellular level (e.g., costimulatory and coinhibitory molecules) to that of a single molecule (e.g., CD4 nonpolar patch and ICD helix). Furthermore, our data concerning the residue covariation and functional coordination of the ectodomain nonpolar patch and intracellular helix provide evidence that a multidomain transmembrane protein, which serves as one component of a multimodule receptor complex, can coevolve binding activity on the outside of a cell with regulatory activity on the inside of the cell to dictate the molecule's function within the multimodule receptor complex. We expect this to emerge as a common theme for other complex transmembrane receptors tasked with relaying information across the membrane to the intracellular signaling machinery. Collectively, our results advance our understanding of T cell biology and have translational value given efforts to engineer synthetic receptors for therapeutic purposes, such as chimeric antigen receptors (CARs). While CAR-T cell therapy has shown considerable promise, problems with sensitivity and side effects now suggest that the absence of mechanisms to mediate or regulate the relay of information across the membrane have a fitness cost for this form of CAR (*Labanieh et al., 2018*). We therefore think that biomimetic designs, incorporating strategies refined over ~435 million years of iterative testing in a variety of vertebrates, will ultimately lead to more sensitive and reliable synthetic receptors (*Kobayashi et al., 2020*). Such biomimetic engineering will require a doubling down on basic research efforts to elucidate the evolutionary blueprint for key immune receptors.

## Materials and methods
### Evolutionary analyses

Available CD4 orthologs were identified through reciprocal blast-based searches and downloaded from GenBank. BLAST may not only identify orthologs, so additional criteria were used to include putative orthologous CD4 sequences in our analyses: the presence of a domain structure consisting of four extracellular Ig domains followed by a TMD and a C-terminal ICD, including the presence of the Lck binding clasp (CxC). Sequences that were shorter, contained frameshift mutations, or displayed high sequence variability were excluded from the analysis. For the current study, teleost fish were considered to be the oldest living species that contain a CD4 molecule given that a CD4 ortholog was not reported in the elephant shark (*Callorhinchus milii*), although future analyses of other cartilaginous fishes might yield more distant orthologs (*Venkatesh et al., 2014*). The final dataset contained 99 unique CD4 orthologs, ranging from teleost fish to human. These (putative) coding sequences were translated to amino acids and aligned using MAFFT (*Katoh et al., 2002*). For codon-based analyses, the aligned amino acid sequences were back translated to nucleotides to maintain codons. The multiple sequence alignments were further processed to remove all insertions (indels) relative to the mouse CD4 sequence (NM_013488.3) to maintain consistent numbering of sites. The 5′ and 3′ regions of the CD4 molecules were not consistently aligned due to different start codon usage or extensions of the ICD, respectively. The alignment was edited to start at the codon (AAG) coding for K48 within mouse CD4. The alignment that includes all 99 CD4 sequences ends at the last cysteine residue that

makes up the CQC clasp. For the mammalian only dataset, the alignment terminates at the mouse CD4 stop codon.

FastTree was used to estimate maximum likelihood trees (*Price et al., 2010*). For amino acid-based trees, the Jones–Taylor–Thornton (JTT) model of evolution was selected, while the general time-reversible model was used for nucleotide trees (*Jones et al., 1992*; *Waddell and Steel, 1997*). In all conditions, a discrete gamma model with 20 rate categories was used. A reduced representation tree (based on the amino acid alignment) is shown in *Figure 1A*. We used the same JTT model to estimate the marginal reconstructions of nodes indicated in *Figure 1A*. Phylogenetic trees and logo plots were visualized in Geneious and further edited using Adobe Illustrator.

Ancestral sequences were estimated using GRASP (*Foley et al., 2020*).

Codon-based analysis of selection was performed using the hypothesis testing FEL model as implemented within the phylogenies (HyPhy) package (version 2.5.14) (MP) (*Kosakovsky Pond et al., 2020*; *Pond et al., 2005*; *Weaver et al., 2018*). The back-translated codon-based alignments described above were used for these analyses. FEL uses likelihood ratio tests to assess a better fit of codons that allowed selection (p < 0.1). When calculating values for all CD4 orthologs included in the initial phylogenetic analysis we analyzed and identified the sequences within the mammalian clade as the foreground branches on which to test for evolutionary selection in order to maximize statistical power.

Covariation between protein residues was calculated using the MISTIC2 server. We calculated four different covariation methods (MIp, mFDCA, plmDCA, and gaussianDCA) (*Colell et al., 2018*). Protein conservation scores were calculated based on the protein alignment using the ConSurf Server (*Ashkenazy et al., 2016*; *Ashkenazy et al., 2010*). ConSurf conservation scores are normalized, so that the average score for all residues is zero, with a standard deviation of one. The lower the score, the more conserved the protein position. For the purpose of this study, residues were considered to covary if the MI was larger than 4 and both residues had a ConSurf conservation score lower than −0.5. Also, pairs with an MI larger than 8 were considered to covary if the conservation score was below −0.3. Using these criteria, we selected 0.5% of all possible pairs as recommended (*Buslje et al., 2009*; *Colell et al., 2018*).

Raw data, including alignments and phylogenetic trees, associated with *Figure 1*, *Supplementary file 1* are available on Dryad (https://doi.org/10.5061/dryad.59zw3r26z).

## Cell lines

58α⁻β⁻ T cell hybridoma lines were generated from Kuhns Lab stocks of parental 58α⁻β⁻ T cell hybridoma cells (obtained from Y.H. Chien at Stanford University) by retroviral transduction and maintained in culture by standard techniques as previously described (*Glassman et al., 2018*; *Letourneur and Malissen, 1989*). 58α⁻β⁻ T cell hybridomas lack expression of endogenous TCRα and TCRβ chains, are CD4 negative, make IL-2 in response to TCR signaling, and are variant of the DO-11.10.7 mouse T cell hybridoma (Balb/c T cell fused to BW5147 thymoma) (*Letourneur and Malissen, 1989*). We validate the cells lines by these characteristics as well as expression of H2-D$^d$ to validate Balb/c origin. In brief, 1 day after transduction the cells were cultured in 5 µg/ml puromycin(Invivogen) and 5 µg/ml zeocin (Thermo Fisher Scientific) in RPMI 1640 (Gibco) supplemented with 5% fetal bovine serum (FBS) (Atlanta Biologicals or Omega Scientific), penicillin–streptomycin–glutamine (Cytiva), 10 µg/ml Ciprofloxacin (Sigma), and 50 µM beta-2-mercaptoethanol (Thermo Fisher Scientific). The next day drug concentrations were increased to 10 µg/ml puromycin (Invivogen) and 100 µg/ml zeocin (Thermo Fisher Scientific) in 10 ml in a T25 flask. Aliquot of $1 \times 10^7$ cells were frozen at days 5, 7, and 9. Cells were thawed from the day 5 freeze and cultured for 3 days in 10 µg/ml puromycin and 100 µg/ml zeocin, and maintained below $1 \times 10^6$ cells/ml to use in the functional assays. Cells used in the functional assays were grown to $0.8 \times 10^6$ cells/ml density and replicates of three functional assays were performed every other day. If cells exceeded $1 \times 10^6$ cells/ml at any point in the process they were discarded as they lose reactivity at high cell densities and a new set of vials was thawed. Typically, two independent WT and mutant pairs were generated for any given mutant and tested for IL-2 to gain further confidence in a response phenotype. When cells lines are presented together in a graph, that indicates that the cell lines (WT and mutants sets) were generated at the same time from the same parental cell stock.

Given the number of mutant CD4 cell lines generated and handled in this study, the identity of the transduced CD4 gene was verified by PCR sequencing at the conclusion of three independent

functional assays. 58α⁻β⁻ T cell hybridomas were lysed using DirectPCR Tail Lysis Buffer (Viagen Biotech) with proteinase K (Sigma) for 2 hr at 65°C. Cells were then heated at 95°C for 10 min. Cell debris was pelleted, and the supernatants were saved. CD4 was amplified by PCR using Q5 DNA Hot Start Polymerase (New England BioLabs) in 0.2 µM dNTP, 0.2 µM primer concentration, Q5 reaction buffer, and water. CD4 was amplified using the following primers:

> 5' primer: acggaattccgctcgagcgccaccatggtgcgagccatctctctcttagg
> 3' primer: ctagcaagcttgtcgactcaagatcttcattagatgagattatggctcttctgc

Product were purified using SpinSmart Nucleic Acid Purification Columns (Thomas Scientific) and sent to Eton Bioscience for sequencing with the following 5' CD4 primer: gtctctgaggagcagaag.

The I-E^{k+} M12 cells used as APCs were previously reported (*Glassman et al., 2018*). M12 cells are a murine B cell lymphoma from Balb/c mice (H2-D^d validated) (*Kim et al., 1979*). Cells were cultured in RPMI 1640 (Gibco) supplemented with 5% FBS (Atlanta Biologicals or Omega Scientific), pencillin–streptomycin–glutamine (Cytiva), 10 µg/ml ciprofloxacin (Sigma), 50 µM beta-2-mercaptoethanol (Thermo Fisher Scientific), and 5 µg/ml puromycin (Invivogen) and 50 µg/ml Zeocin (Thermo Fisher). The parental cells are maintained in Kuhns Lab stocks and were originally obtained from MM Davis stocks (Stanford University).

Parental 58α⁻β⁻ T cell hybridoma and M12 cells are periodically treated with Plasmocyn and tested for mycoplasma contamination by PCR using primer sequences available from American Type Culture Collection (ATCC) (5' primer sequence: TGCACCATCTGTCACTCTGTTAACCTC; 3' primer sequence: GGGAGCAAACAGGATTAGATACCCT). All transduced cell lines used here were grown in ciprofloxacin and generated from parental cells that had tested negative for mycoplasma.

For retroviral production we used Phoenix-eco cells from the Nolan Lab (ATCC CRL-3214).

## Antibodies

| Antibodies | Vendor | Catalog number | RRID |
|---|---|---|---|
| anti-mouse CD4 eFlour 450, clone GK1.5 | Thermo Fisher Scientific | 48-0041-82 | AB_10718983 |
| anti-mouse TCRα APC, clone RR8-1 | Thermo Fisher Scientific | 17-5800-82 | AB_19853170 |
| anti-mouse CD3ε PE-Cy7, clone 145–2 C11 | Thermo Fisher Scientific | 25-0031-82 | AB_469572 |
| anti-mouse IL-2, clone JES6-1A12 | BioLegend | 503,702 | AB_315292 |
| biotin anti-mouse IL-2, clone JES6-5H4 | BioLegend | 503,804 | AB_315298 |
| Streptavidin HRP | BioLegend | 405,210 | |
| anti-mouse TCRβ PE, clone KJ25 | BD Pharmingen | 553,209 | AB_394709 |
| biotin anti-mouse CD4 (Clone RM4-4) | BioLegend | 116,010 | AB_2561504 |
| anti-mouse CD4 APC, clone GK1.5 | BioLegend | 100,412 | AB_312697 |
| anti-mouse Lck PE, clone 3A5 | Santa Cruz | sc-433 | |
| Cholera Toxin Subunit B Alexa Fluor 488 | Thermo Fisher Scientific | C22841 | |
| anti-mouse pCD3 ζ Alexa Flour 647, clone K25-407.69 | BD Phosflow | 558,489 | AB_647152 |
| anti-mouse pZap70 APC, clone n3kobu5 | Thermo Fisher Scientific | 17-9006-42 | AB_2573268 |
| anti-mouse pPlcγ1 PE, clone A17025A | BioLegend | 612,404 | AB_2801120 |

## Flow cytometry

Cell surface expression of CD4 and TCR–CD3 complexes was measured by flow cytometry. In brief, cells were stained for 30 min at 4°C in Fluorescence-Activated Cell Sorting (FACS) buffer (phosphate-buffered saline [PBS], 2% FBS, and 0.02% sodium azide) using anti-CD4 (clone GK1.5, eFluor 450 conjugate, Thermo Fisher Scientific), anti-TCRα (anti-Vα11, clone RR8-1, APC conjugate, Thermo Fisher Scientific), anti-CD3ε (145-2C11, Thermo Fisher Scientific), and GFP was detected as a measure of the TCRβ-GFP subunit. Analysis was performed on a Canto II or LSRII (BD Biosciences) at the Flow

Cytometry Shared Resource at the University of Arizona. Flow cytometry data were analyzed with FlowJo Version 9 software (Becton, Dickinson & Company).

## Functional assays

IL-2 production was measured to quantify pMHCII responses. $5 \times 10^4$ transduced $58\alpha^-\beta^-$ T cell hybridomas were cocultured with $1 \times 10^5$ transduced I-E$^{k+}$ M12 cells in triplicate in a 96 well round bottom plate in RPMI with 5% FBS (Omega Scientific), Pen-Strep + L-glutamine (Cytiva), 10 ng/ml ciprofloxacin (Sigma), and 50 µM beta-2-mercaptoethanol (Fisher) in the presence of titrating amounts of MCC 88–103 peptide (purchased from 21st Century Biochemicals at >95% purity) starting at 30 µM MCC and a 1:3 titration (*Glassman et al., 2018*). For experiments with APCs expressing tethered pMHCII, $5 \times 10^4$ $58\alpha^-\beta^-$ T cell hybridomas were cultured with $1 \times 10^5$ MCC:I-E$^{k+}$ or T102S:I-E$^{k+}$ M12 cells in triplicate in a 96-well round bottom plate using the same culture conditions as above. The supernatants were collected and assayed for IL-2 concentration by ELISA after 16 hr of co-culture at 37°C. Anti-mouse IL-2 (clone JES6-1A12, BioLegend) antibody was used to capture IL-2 from the supernatants, and biotin anti-mouse IL-2 (clone JES6-5H4, BioLegend) antibody was used as the secondary antibody. Streptavidin–Horse Radish Peroxidase (HRP) (BioLegend) and 3,3',5,5'-Tetramethylbenzidine (TMB) substrate (BioLegend) were also used.

To assess engagement-induced endocytosis, CD4 surface levels were measured by flow cytometry 16 hr after coculture with APCs and peptide as described above for IL-2 quantification. 96-Well plates containing cells were washed with ice cold FACS buffer (PBS, 2%FBS, 0.02% sodium azide), transferred to ice, and Fc receptors were blocked with Fc block mAb clone 2.4G2 for 15 min at 4°C prior to surface staining for 30 min at 4°C with anti-CD4 (clone GK1.5 EF450, Invitrogen) and anti-Vβ3 TCR clone (clone KJ25, BD Pharmingen) antibodies. Cells were washed with FACS buffer prior to analysis on a LSRII (BD Biosciences) at the Flow Cytometry Shared Resource at the University of Arizona. Flow cytometry data were analyzed with FlowJo Version 10 software (Becton, Dickinson & Company). The average of the geometric mean of the TCR or CD4 signal was taken for the triplicate of the post $58\alpha^-\beta^-$ cells cocultured with M12 I-E$^{k+}$ cells at 0 µM MCC concentration. Each value of the raw gMFI of TCR or CD4 for cells cultured at 10 µM MCC was subtracted from the average gMFI at 0 µM. The values show the change of gMFI from 0 to 10µM.

For ELISpot analysis, $1.25 \times 10^3$ transduced $58\alpha^-\beta^-$ T cell hybridomas were mixed with $1.5 \times 10^5$ M12 cells that expressed MCC peptide tethered to I-E$^k$ in triplicate wells on a mixed cellulose ester membrane plate (Merck Millipore) coated with 10 µg/ml anti-mouse IL-2 (clone JES6-1A12, BioLegend) antibody. Cells were co-cultured for 16 hr at 37°C in culture media as listed above. Plates were washed, probed with biotin anti-mouse IL-2 (clone JES6-5H4, BioLegend), washed, and probed with streptavidin–HRP (BioLegend). KPL TrueBlue Peroxidase Substrate (Sera Care) were used to identify spots according to the manufacturer's instructions. Spots and spot intensity were enumerated from triplicate wells on a ImmunoSpot counter from Cellular Technologies Limited using the ImmunoSpot 7.0.13.0 software.

## Sucrose gradient analysis

Membrane fractionation by sucrose gradient was performed similar to previously described methods (*Hur et al., 2003*; *Parrish et al., 2016*). For cell lysis, $6 \times 10^7$ $58\alpha^-\beta^-$ T cell hybridomas were harvested and washed 2× using TNE buffer (25 mM Tris, 150 mM NaCl, 5 mM Ethylenediaminetetraacetic acid (EDTA)). Cells were lysed on ice in 1% Triton-X detergent in TNE in a total volume of 1 ml for 10 min and then dounce homogenized 10×. The homogenized lysate was transferred to 14 × 95 mm Ultraclear Ultra Centrifuge tubes (Beckman). The dounce homogenizer was rinsed with 1.6 ml of the 1% lysis buffer, which was then was added to the Ultracentrifuge tube. 2.5 ml of 80% sucrose was added to the centrifuge tube with lysate and mixed well. Gently, 5 ml of 30% sucrose was added to the centrifuge tubes, creating a 30% sucrose layer above the ~40% sucrose/lysate mixture. Then, 3 ml of 5% sucrose was added gently to the centrifuge tube, creating another layer. The centrifuge tubes were spun 18 hr at 4°C in a SW40Ti rotor at 36,000 rpm.

Analysis of membrane fractions was performed via flow-based fluorophore-linked immunosorbent assay (FFLISA) as previously described (*Parrish et al., 2016*). In brief, 88 µl of Streptavidin Microspheres 6.0 µm (Polysciences) were coated overnight at 4°C with 8 µg of biotinylated anti-CD4 antibody (clone RM4-4, BioLegend). Prior to immunoprecipitation, beads were washed with 10 ml of

FACS buffer (1× PBS, 2%FBS, 0.02% sodium azide) and resuspended in 3.5 ml of 0.1% Triton X-100 lysis wash buffer in TNE. For each cell line lysed, 10 FACS tubes were prepared with 50 µl of the washed RM4-4 coated beads. Upon completion of the spin, 500 µl was carefully taken off the top of the centrifuge tubes and discarded. Following this, 1 ml was extracted from the top of the tube, carefully as to not disrupt the gradient, and added to a FACS tube with coated beads and capped. This was repeated for 10 individual fractions in separate FACS tubes. Following the extraction, lysates were incubated with the beads for 90 min, inverting the tubes to mix every 15 min.

Following the immunoprecipitation, FACS tubes were washed 3× using 0.1% Triton-X lysis wash buffer in TNE. Tubes were then stained using 1 µl anti-CD4 (APC conjugate; clone GK1.5, BioLegend), 1.5 µl anti-Lck (PE conjugate, clone 3A5, Santa Cruz Biotechnology), and 1 µl CTxB (AF 488 conjugate, Thermo Fisher Scientific, resuspended as per manufacturer's instructions) for 45 min at 4°C. Following the stain, tubes were washed using 0.1% Triton-X lysis wash buffer in TNE. Analysis of beads was performed on a LSRII (BD Biosciences) at the Flow Cytometry Shared Resource at the University of Arizona. $10^4$ events were collected per sample. Flow cytometry data were analyzed with FlowJo Version 10 software (Becton, Dickinson & Company).

For FFLISA analysis, raw gMFI values for fraction 1 were subtracted from the rest of the fractions to account for background, such that the gMFI of fraction 1 is 0. To normalize the data, the percentage of CD4 within any given fraction (fx) relative to the total CD4 gMFI (CD4 signal % of total) was calculated by dividing the gMFI signal in a given fraction (fx) by the sum of the total CD4 gMFI signal [sum(f1:f10)CD4 gMFI] and multiplying by 100 [e.g., fx % of total = fxCD4 gMFI/ Sum(f1:f10)CD4 gMFI × 100]. To normalize the CTxB and Lck signal in any given fraction relative to the CD4 signal in that same fraction (CTxB or Lck normalized to CD4) the gMFI of CTxB or Lck in fx was divided by the CD4 gMFI of that fx and then multiplied by the percentage of CD4 within fx (e.g., Normalized fx Lck = fx Lck gMFI/fx CD4 gMFI × fx CD4% of total CD4 gMFI). AUC analysis was performed with GraphPad Prism 9 for fractions 1–6 to determine the AUC for the DRM domains due to their floating phenotypes, and for fractions 6–10 to determine the AUC for the DSM domains.

## Intracellular signaling analysis

M12 cells expressing Hb:I-E$^k$ (null) or MCC:I-E$^k$ (cognate) tethered pMHCII complexes were labeled with Tag-it Violet according to the manufacturer's instructions (BioLegend). M12 cells and 58α$^-$β$^-$ cells were then chilled on ice for 30 min, $5 \times 10^5$ of each cell type were mixed together in 1.5 ml snap cap tubes, and the cells were pelleted at 2000 rpm for 30 s at 4°C to force interactions. The supernatant was removed and the tubes were transferred to a 37°C water bath for 2 min to enable signaling. Fixation Buffer (BioLegend Inc) was then added for 15 min at 37°C. Cells were washed twice with FACs buffer, pelleted at 350 × g for 5 min at room temperature, resuspended in 1 ml True-Phos Perm Buffer (BioLegend Inc), and incubated at −20°C for 16 hr.

Cells were blocked with anti-mouse FcRII mAb clone 2.4G2 hybridoma supernatants (ATCC) for 30 min, pelleted, and stained on ice for 60 min with anti-pCD3 ζ (clone K25-407.69, Alexa Fluor 647 conjugate, BD Biosciences) in one sample tube, or with anti-pZap70 (clone n3kobu5, APC conjugate, Invitrogen) and anti-pPlcγ1 (clone A17025A, PE conjugate, BioLegend) in a separate sample tube at the vendor-recommended concentrations. Finally, cells were washed 2× with FACS buffer at 1000 × g for 5 min at room temperature and analyzed on a Canto II (BD Biosciences) at the Flow Cytometry Shared Resource at the University of Arizona or on a BD Fortessa. $1 \times 10^4$ 58α$^-$β$^-$ cell:M12 cell couples were collected per sample.

Flow cytometry data were analyzed with FlowJo Verison 10 software (Becton, Dickinson & Company) by gating on 58α$^-$β$^-$ and M12 cell couples, as described previously (*Glassman et al., 2018*). Histograms of the pCD3 ζ, pZap70, or pPlcγ1 intensity for the gated population were then generated and data expressing the gated populations as numbers of cells within intensity bins was exported from FlowJo into Microsoft Excel where the number of cells for each bin intensity value for MCC:I-E$^k$ stimulated cells was subtracted from Hb:I-E$^k$ stimulated cells on a bin-by-bin basis. This allowed us to enumerate the intensity differences per bin upon stimulation with the agonist pMHCII over background. Mean intensity and standard error of the mean were calculated based on the background subtracted (MCC:I-E$^k$-Hb:I-E$^k$) data. The data were then transferred to Prism 9 where we performed smoothing analysis with 500 nearest neighbors to smooth the line profile for graphing purposes.

Those intensity bins with positive values were considered to contain cells that had responded to the MCC:I-E$^k$ stimuli above background.

## Statistical analysis

Statistical analyses of sucrose gradient and functional assays were performed with GraphPad Prism 9 software as indicated in the figure legends. For each functional assay (IL-2 production and CD4 endocytosis), each individual experiment (biological replicate) was performed with triplicate analysis (technical replicates) and each experiment was repeated three times (three biological replicates). For sucrose gradient analysis, 10$^4$ beads were collected by flow cytometry in each experiment (technical replicates) and each experiment was performed three times (biological replicates). Three biological replicates were chosen for each analysis as per convention, and no power calculations were determined. One-way analysis of variance was performed with a Dunnett's posttest when all mutants tested in an experiment were compared to a control sample (e.g., WT). Sidak's posttest were applied when comparing between two specific samples. These posttests were chosen based on Prism recommendations. Student's unpaired t-tests (two-tailed) were performed when comparing WT and LL mutant samples only for phosphorylation analysis.

## Materials availability statement

Raw data, including alignments and phylogenetic trees, associated with *Figure 1*, *Supplementary file 1* are available on Dryad (https://doi.org/10.5061/dryad.59zw3r26z). Cell lines and constructs are available upon request.

# Acknowledgements

This work was supported by the National Institutes of Health/National Institute of Allergy and Infections Diseases Grant R01AI101053 (MSK), the Pew Scholars Program in the Biomedical Sciences (MSK), the Cancer Center Support Grant CCSG-CA 023074 for flow cytometry (MSK), and AZ TRIF funds from the BIO5 Institute (KVD). We thank Thomas Serwold and Leslie Berg for critical feedback as well as Dominik Schenten, Caleb Glassman, Joseph Harrison, Piet Maes, and Benjamin Renquist for critically reading the manuscript.

# Additional information

### Competing interests

Michael S Kuhns: has disclosed an outside interest in Module Therapeutics to the University of Arizona. Conflicts of interest resulting from this interest are being managed by the University of Arizona in accordance with their policies. The other authors declare that no competing interests exist.

### Funding

| Funder | Grant reference number | Author |
|---|---|---|
| National Institute of Allergy and Infectious Diseases | R01AI101053 | Michael S Kuhns |
| Cancer Center Support Grant | CCSG-CA 023074 | Michael S Kuhns |
| AZ TRIF Funds | | Koenraad Van Doorslaer |

The funders had no role in study design, data collection, and interpretation, or the decision to submit the work for publication.

### Author contributions

Mark S Lee, Resources, Data curation, Formal analysis, Validation, Investigation, Visualization, Methodology, Writing – review and editing; Peter J Tuohy, Data curation, Formal analysis, Validation, Investigation, Writing – review and editing; Caleb Y Kim, Data curation, Formal analysis, Validation, Investigation, Visualization, Methodology, Writing – review and editing; Katrina Lichauco, Investigation,

Writing – review and editing; Heather L Parrish, Conceptualization, Resources, Validation, Investigation, Methodology, Writing – review and editing; Koenraad Van Doorslaer, Conceptualization, Resources, Data curation, Formal analysis, Validation, Investigation, Visualization, Methodology, Writing – original draft; Michael S Kuhns, Conceptualization, Resources, Data curation, Formal analysis, Supervision, Funding acquisition, Validation, Investigation, Visualization, Methodology, Writing – original draft, Project administration, Writing – review and editing

### Author ORCIDs
Katrina Lichauco ⓘ http://orcid.org/0000-0002-9480-2893
Koenraad Van Doorslaer ⓘ http://orcid.org/0000-0002-2985-0733
Michael S Kuhns ⓘ http://orcid.org/0000-0002-0403-6313

### Decision letter and Author response
Decision letter https://doi.org/10.7554/eLife.79508.sa1
Author response https://doi.org/10.7554/eLife.79508.sa2

## Additional files

### Supplementary files
• MDAR checklist
• Supplementary file 1. Table 2.
• Supplementary file 2. Construct sequences.

### Data availability
Raw data, including alignments and phylogenetic trees, associated with figures 1 and S1 as well as source data and statistics for remaining figures are available on Dryad (https://doi.org/10.5061/dryad.59zw3r26z).

The following dataset was generated:

| Author(s) | Year | Dataset title | Dataset URL | Database and Identifier |
|---|---|---|---|---|
| Lee M, Tuohy P, Kim C, Lichauco K, Parrish H, Van Doorslaer K, Kuhns M | 2022 | Data associated with "Enhancing and inhibitory motifs have coevolved to regulate CD4 activity" | https://doi.org/10.5061/dryad.59zw3r26z | Dryad Digital Repository, 10.5061/dryad.59zw3r26z |

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
