## [Editor Report]

This paper takes an evolutionary approach to investigate the mechanisms by which CD4 regulates T-cell receptor activation and downstream functional responses. The authors identify conserved and coevolving motifs in the extracellular, transmembrane, and intracellular domains of CD4 that appear to regulate multiple aspects of its function. These findings suggest a recalibration of the perception of CD4 as simply an accessory to the central complex of T-cell receptors and CD3 in pMHC-specific T-cell responses.

---

## [Decision Letter]

**Decision letter after peer review:**

[Editors’ note: the authors submitted for reconsideration following the decision after peer review. What follows is the decision letter after the first round of review.]

Thank you for submitting the paper "Enhancing and inhibitory motifs have coevolved to regulate CD4 activity" for consideration at *eLife*. Your article has been reviewed by 3 peer reviewers, and the evaluation has been overseen by a Senior Editor. Although the work is of interest, we regret to inform you that the findings at this stage are too preliminary for further consideration at *eLife*.

All three reviewers agreed that the comparative genomics/evolution-motivated approach to identify new CD4 functions is intriguing and likely successfully identified 'novel' regions of CD4 that contribute to its function. However, there was also agreement that the major findings of the manuscript, while intriguing, fail to meet the significance bar for *eLife* due largely to a lack of mechanistic details of how distal regions of CD4 interact (thermodynamically or functionally) and control downstream TCR activation. In addition, the reviewers commented that the manuscript is written in a way that is largely inaccessible to non-experts that comprise much of the *eLife* readership and, as such, the broad and generalizable impact of the work was not apparent.

*Reviewer #1 (Recommendations for the authors):*

Lee and colleagues use evolutionary genomics to identify regions of CD4 that appear to have evolved under functional constraint and which were not previously appreciated to contribute to CD4 function. They proceed to test the function of these regions using in vitro assays to measure how mutations in these regions affect several aspects of CD4 biology including changes in CD4 localization on the membrane, interaction of CD4 with the downstream kinase Lck, and ultimately IL-2 production. The authors find effects on IL-2 production from mutations in all of identified motifs. Of interest, mutations in three of the regions tend to decrease IL-2 production, but mutations in one motif of the intracellular domain (termed "helix" or "IKRLL") increase IL-2 production. From this data, the authors also find evidence that interaction with the Lck kinase may not be necessary to stimulate IL-2 production, counter to the established model of CD4 signaling. Next, an analysis of statistical covariation in CD4 protein sequences showed significant covariation between the intracellular helix and most of the newly identified functional motifs in the intracellular, transmembrane, and extracellular domains. Finally, the authors combine mutations in the intracellular helix (which increase IL-2; "LL") with mutations in evolutionarily covarying regions (which decrease IL-2; "R426A, Clasp, TP") to measure the effects of the double mutants relative to each single mutant. All double mutants appear to behave as the additive effects of the individual mutations, without clear non-additive or epistatic effects.

Overall, the authors use a powerful combination of evolutionary analyses and biochemistry/molecular biology to identify novel functional regions of CD4. However, the author's conclusion that these motifs have coevolved to "finetune" the MHCII response are overstated and not supported by the functional data. Specifically, it is unclear what evidence the authors provide to support regulation of the ECD by intracellular or transmembrane regions. In figure 7, R426A, clasp, and TM mutants all decrease IL-2 on their own. The effects of these mutations in the background of the LL mutation seem to be simply additive, providing no evidence of interaction between these motifs (a la a network of coevolving residues). The covariation could reflect a 'rheostat' wherein mutations in a stimulatory domain are compensated by mutations in an enhancing domain and vice versa, but there need be no regulation in the sense of a perturbation/input at one motif affects the function of another/allostery.

To be suitable for a broad readership journal like *eLife*, the manuscript would benefit from a figure which shows the biological context (all the relevant molecules and the regions of interest in CD4). In addition, the diagrams included in the supplementary showing the mutations and their location within CD4 could be included in this or should be moved to the main figures. Similarly, some summary figure showing the mutations and their effects in a simple manner (up/down/wt) would be immensely useful to process the proposed interaction of mutations.

The relevance of the functions measured is not clear – what do we know about the function of raft vs non-raft CD4? Most CD4 is in DSM/non-raft fraction. Why use CTxB and fractionation? Do they tell different things? What about when they don't agree? A clear setup of the why these assays have been chosen is crucial to understanding the relevance of the data.

The language should be more precise regarding evolutionary inferences, with a clear delineation of what is a statistical test and what is an inference of those tests. For example:

– The pairwise conservation/mutual information analysis detects statistical covariation which suggests coevolution. This analysis does not show coevolution.

– They find statistical signatures of selection, not identify selection.

– (ln 104-105) "how ancient and ongoing environmental challenges have influenced CD4" should read "find motifs that have been preserved".

The various monikers used for each motif are hard to keep track of. It would greatly help the reader to have a single term to apply to each motif in the text and figures (e.g. Palm and CV+C, ECD and Domain 3).

*Reviewer #2 (Recommendations for the authors):*

"Enhancing and inhibitory motifs have coevolved to regulate CD4 activity" takes an evolutionary approach to investigate the mechanism(s) by which CD4 regulates TCR activation and downstream functional responses. The authors identify conserved motifs in the extracellular, transmembrane, and intracellular domains of CD4 that appear to regulate multiple aspects of its function, including its localization to lipid RAFTs, its capacity to interact with Lck, and its ability to promote IL-2 production. Notably, one of these motifs, comprising a helix just N-terminal to the cysteine clasp responsible for interaction with Lck, has an inhibitory effect on TCR signaling, and it seems to have coevolved in eutherians together with a motif in extracellular domain 3 that promotes T cell activation. Collectively, these results suggest that the regulation of CD4 activity has been finely tuned during evolution such that the acquisition of activity-promoting mutations is balanced by the emergence of inhibitory regions. Although this idea is interesting, it should be accompanied by more in-depth mechanistic work to demonstrate exactly how specific parts of CD4 control TCR activation. The notion that motifs within CD4 have coevolved is not particularly unexpected. These evolutionary relationships should be connected to mechanistic and functional insights into how the molecule works.

1) Just because motifs have coevolved and their mutations have additive effects doesn't necessarily mean that they are functionally coupled. Mechanistic insight will require a more in-depth analysis of membrane proximal signaling events and the activation status of downstream pathways (e.g. Erk, Ca, NFKB).

2) The idea that the role of CD4 is to control the colocalization of the TCR and Lck to RAFTs is quite interesting, and it should be tested. Could RAFT localization be modulated independently of CD4, and would this reverse the effects of the relevant CD4 mutations?

3) It is difficult to get a sense of what the fractionation and lipid association results really mean in terms of membrane organization (e.g. Figure 2). CTxB binding in the context of a bead-based IP is particularly unphysiological. A positive control would be helpful here. What would an established disruption of RAFT localization look like in this assay (e.g. a mutation in the Lck N-terminal domain).

4) Based on the presented data, one might conclude that mutation of the ICD helix enhances T cell responses simply by inhibiting CD4 internalization, thereby maintaining the density of CD4 on the cell surface. Indeed, all of the mutations that enhance IL-2 responses appear to result in higher surface expression. Do the authors favor this hypothesis, or can it be ruled out?

5) Given the evolutionary focus of this study, it would have been more interesting if the authors had actually used sequences from non-eutherian CD4s, as opposed to Ala and Gly substitutions. Was this attempted?

6) It is hypothesized that the importance of the GGXXG motif may depend on the local cholesterol concentration of the membrane domain in which it resides. Does changing membrane cholesterol modulate the effects of mutations in the GGXXG motif?

7) The authors should confirm that the CV+C mutation actually has the intended consequence of altering CD4 palmitoylation.

*Reviewer #3 (Recommendations for the authors):*

In this manuscript, Lee et al., the authors interrogate the function of CD4 from an evolutionary perspective. Adaptive immunity within the jawed vertebrates is believed to have arisen as a consequence of the "Immunological Big Bang". This event marks the evolutionary origin of both T and B cell receptors, as well as many of the molecules that mediate their signaling (e.g., the TCR co-receptors CD4 and CD8, as well as the MHC I and II proteins). Although orthologs of TCRs, CD4, and MHCII are broadly conserved throughout jawed vertebrates, these proteins have not yet been identified outside of this lineage. Notably, all three molecules first appear within the cartilaginous fishes, suggesting that their functions may be tightly linked. However, most studies on TCR function focus specifically on TCRs interacting directly with either CD4 or MHCII. In this study, the authors draw on evolutionary analyses to identify broadly conserved sequence motifs that may be involved in direct interactions between CD4 and MHCII to fine-tune the strength of TCR signaling. They complement these computational analyses with biochemical studies to support their hypothesis.

The primary strength of this manuscript lies in the author’s integrative approach; their evolutionary hypotheses are considerably strengthened by the findings from the experimental techniques. However, in its current form, the manuscript is very difficult to follow for a non-specialist in TCR signaling. The authors use a considerable amount of jargon and abbreviations. Consequently, the logic underlying their experiments is not always clear. Thus, while their biological findings may be of interest to researchers interested in the specifics of TCR signaling (particularly from the standpoints of engineering biomimetics and clinically manipulating TCR signaling strength), it is difficult to extend the significance of these findings more broadly.

As mentioned above, the manuscript contains a nearly impenetrable amount of abbreviations and jargon. I suggest that the authors have the manuscript revised by someone outside the field who is familiar with the work to address this problem. This will make the manuscript more considerably more accessible to a broad range of biologists.

Two additional points that should be addressed:

1. On line 55, the authors state that "[in sharks]… an orthologous gene encoding CD4

appears to be absent, as are genes for proteins associated with CD4^+^ 56 T cell helper (Th) or regulatory (Treg) functions (e.g. FoxP3 and Rorc) (Venkatesh et al., 2014)" Although this was the primary finding described in the original elephant shark genome paper, subsequent analysis of this genome (JM Dijkstra, Nature 511, 2014) revealed that the CD4^+^ T cell lineage is likely present within cartilaginous fish.

2. The manuscript relies heavily on evolutionary analysis of CD4 molecules collected from jawed vertebrates. However, the authors do not adequately describe if/how they verified the homology of these molecules. The authors state that "The criteria used for including orthologous CD4 sequences in our analyses were that they have a domain structure consisting of four extracellular Ig domains followed by a TMD and a C-terminal ICD". Many proteins have convergently evolved this domain architecture, which makes this criteria insufficient to assign orthology. A phylogenetic approach is required to confidently determine whether or not the sequences are truly orthologous to CD4. Additionally, all accession numbers and/or aligned sequences should be made freely available in the supplemental material.

[Editors’ note: further revisions were suggested prior to acceptance, as described below.]

Thank you for resubmitting your work entitled "Enhancing and inhibitory motifs regulate CD4 activity." for further consideration by *eLife*. Your revised article has been evaluated by Carla Rothlin (Senior Editor) and a Reviewing Editor.

The manuscript has been improved but there are some remaining issues that need to be addressed, as outlined below:

1) The reviewers feel that the new signaling data are conceptually the right experiments but may be technically limited by overstimulating T cells and sparsely sampling antigen concentrations in some assays. As discussed by Reviewer 3, despite the sparse dose response applied in the signaling experiments, the manuscript presents the conclusions as if there can be only one interpretation of the results.

For example, whether or not the Clasp plays a positive or negative role in TCR signaling is not settled science (which is why your manuscript is interesting). The reviewers don't feel like the added experiments completely resolve these questions, so we request that your either add additional experiments or appropriately temper the presentation and discussion of your conclusions to acknowledge the remaining gaps in our understanding of this important topic.

2) We understand the difficulty of crafting a narrative that is accessible to all readers but still respects the conventions and standards of both immunology and molecular evolution. While we appreciate the revisions made to improve the readability of the manuscript, we respectfully request that you have a colleague outside the evolution and immunology read the manuscript and highlight sections or terminology or acronyms that could be modified with the goal of increasing the accessibility of the narrative. The extent and substance of these revisions are left to your discretion.

*Reviewer #1 (Recommendations for the authors):*

Lee and colleagues use phylogenetics and amino acid conservation/covariation to identify previously unstudied motifs within CD4 that emerged in the last common ancestor of eutherians, statistically coevolved, and have contrasting effects on downstream signaling when mutated. Historically, the activity of CD4 on TCRs has been shown to be indirect – mediated by CD4 binding Lck and Lck phosphorylating CD3. The data in this manuscript (based on mutations in positions of CD4 shown by the authors to coevolve) suggest that CD4 also directly regulates or uses an unknown cofactor (not Lck) to regulate TCR signaling.

The revisions by the authors have significantly improved the manuscript by trimming potentially unsupported assertions about the interaction of distal patches on CD4 and addressing previous concerns regarding interpretation of the evolutionary analyses.. The added experiments to test the effects of various mutants on downstream signaling steps provide a limited glimpse of how the different regions of CD4 may be differentially regulating TCR signaling.

My main concern is still whether the very detailed and rigorous analysis of the interactions among CD4-Lck-CD3-TCR and the resulting findings are accessible enough for a broad readership to understand the potential impact of the finding. The evolutionary analyses are interesting, but not especially novel, so I think the impact of this paper is best judged in the context of its contribution to the CD4/TCR literature.

*Reviewer #2 (Recommendations for the authors):*In their manuscript, "Enhancing and inhibitory motifs regulate CD4 activity", Lee et al., employ an evolutionary approach to identify specific residues within CD4 that impact TCR signaling. CD4 is a transmembrane protein composed of four immunoglobulin domains that mediates T cell signaling in two ways: first, through direct interactions with MHC II on antigen presenting cells; and second, by facilitating intracellular signaling by interacting with the Src kinase, Lck. Traditionally, the researchers have focused on the TCR-MHC II interactions as the key determinants of T cell signaling. However, recent evidence suggests that CD4 may also regulate the outcome of T cell activation. In this study, the authors use phylogenetic analysis as a relatively unbiased means of defining residues within CD4 that are important for signaling. This is predicated on the assumption that protein evolution is constrained by function; residues that directly mediate signaling are less likely to mutate over time. CD4 is a central component of the adaptive immune system and its function is limited to this context. Accordingly, orthologs of CD4 have only been identified within vertebrates.In this study, the authors use computational analyses to characterize the evolution of CD4 within the jawed vertebrates as a foundation for functional experiments. Overall, the manuscript is well-written and the conclusions are supported by the data shown. This work could potentially be of interest to a broad group of researchers, including immunologists interested in adaptive immunity and comparative immunologists as well as cell biologists who are focused on transmembrane cell signaling proteins.

My primary concern with the manuscript is that, in its current form, it is difficult to interpret by a non-specialist in TCR signaling. The manuscript contains a considerable amount of jargon, abbreviations and specific details without the necessary context. I would suggest having the manuscript read by someone who is not familiar with the work and can highlight the more challenging sections.

In a similar vein, the figures are quite complex. The authors might consider simplifying to emphasize the main points.

Specific points to address are outlined below.

1. In the Introduction, the authors refer to the "immunological 'Big Bang' that gave rise to RAG-based antigen receptor gene rearrangement in jawed vertebrates…". This discounts several important advances in the evolution of adaptive immunity, including the characterization of the VLR-based adaptive immune system that evolved in parallel within the jawless vertebrates as well as the discovery of orthologs of the RAG1/2 and CDA enzymes within invertebrate deuterostomes. These findings, which are described and synthesized in Flajnik, 2014 (doi:10.1016/j.cub.2014.09.070), will provide important evolutionary context for this manuscript.

2. Line 69. The authors describe MHC-TCR signaling as mediated by "5 distinct modules", but it is not clear from the subsequent text what these modules are. This should be clarified.

3. Although the authors describe MHC-TCR signaling in great detail, the structure of CD4 is not described. This would be particularly useful to readers who are not specialists in T cell biology and might include a figure showing the four Ig domains, the transmembrane region, and the intracellular domain. The abbreviations ECD, TMD, and ICD should be clearly defined early in the text.

4. Line 115. The authors should briefly mention in the text which CD4 orthologs were used. For example, "X CD4 orthologs were analyzed from Y species, which included the vertebrate lineages …". This should be further elaborated in the methods section. The authors state (line 682) that "… available CD4 orthologs were downloaded from Genbank." More details are necessary. There are many transmembrane proteins that consist of four Ig domains; only a subset of these have been correctly annotated as CD4. The authors should specifically describe their inclusion/exclusion criteria (i.e., define the algorithms and parameters used and the "etc" on line 685). Accession numbers for all these sequences should be included in the manuscript.

5. Although most mammals have a single ortholog of CD4, due to whole genome duplications, many teleost genomes contain two paralogs (e.g., doi:10.4049/jimmunol.1600222). Were these included in the analysis? Conversely, the authors cite Venkatesh et al., 2014 as evidence of the absence of CD4 in sharks. This manuscript was followed by a comment (doi:10.1038/nature13446) that identified many of the "missing" components of adaptive immunity from the elephant shark genome sequence. It would be wise to double-check these findings.

6. Line 131. It is not clear how the distribution of residues subject to purifying selection suggests that "mutation putative linear motifs within the ICR is selected against". This should be clarified or justified.

*Reviewer #3 (Recommendations for the authors):*

This paper takes an evolutionary approach to investigate the mechanism(s) by which CD4 regulates TCR activation and downstream functional responses. The authors identify conserved motifs in the extracellular, transmembrane, and intracellular domains of CD4 that appear to regulate multiple aspects of its function, including its localization to lipid RAFTs, its capacity to interact with Lck, and its ability to elicit membrane proximal signaling and promote IL-2 production. Notably, one of these motifs, comprising a helix just N-terminal to the cysteine clasp responsible for interaction with Lck, has an inhibitory effect on TCR signaling, and it seems to have coevolved in eutherians together with a motif in extracellular domain 3 that promotes T cell activation. Collectively, these results suggest that the regulation of CD4 activity has been finely tuned during evolution such that the acquisition of activity-promoting mutations is balanced by the emergence of inhibitory regions.

The revisions and new data have improved this manuscript, although I still have some concerns.

1) The manuscript remains a difficult read. I'm not sure how best to deal with this. The acronyms get a bit overwhelming. Perhaps it would be better to remove some of the less used acronyms (e.g. MRCA) and just write the words out in full?

2) The new signaling data (Figure 6) are a welcome addition to the manuscript, but they are rather sparse. In particular, I am worried that the authors may be missing important phenotypes by overstimulating the TCR expressing cells. A dose response should be performed for each signaling readout. It's possible, for instance, that at a lower antigen dose they would observe more substantial differences in CD3z phosphorylation in Figure 6A. This would be consistent with the idea that CD4-Lck interaction matters more when antigenic peptide is limiting.

---

## [Author Response]

[Editors’ note: the authors resubmitted a revised version of the paper for consideration. What follows is the authors’ response to the first round of review.]

Reviewer #1 (Recommendations for the authors):Lee and colleagues use evolutionary genomics to identify regions of CD4 that appear to have evolved under functional constraint and which were not previously appreciated to contribute to CD4 function. They proceed to test the function of these regions using in vitro assays to measure how mutations in these regions affect several aspects of CD4 biology including changes in CD4 localization on the membrane, interaction of CD4 with the downstream kinase Lck, and ultimately IL-2 production. The authors find effects on IL-2 production from mutations in all of identified motifs. Of interest, mutations in three of the regions tend to decrease IL-2 production, but mutations in one motif of the intracellular domain (termed "helix" or "IKRLL") increase IL-2 production. From this data, the authors also find evidence that interaction with the Lck kinase may not be necessary to stimulate IL-2 production, counter to the established model of CD4 signaling. Next, an analysis of statistical covariation in CD4 protein sequences showed significant covariation between the intracellular helix and most of the newly identified functional motifs in the intracellular, transmembrane, and extracellular domains. Finally, the authors combine mutations in the intracellular helix (which increase IL-2; "LL") with mutations in evolutionarily covarying regions (which decrease IL-2; "R426A, Clasp, TP") to measure the effects of the double mutants relative to each single mutant. All double mutants appear to behave as the additive effects of the individual mutations, without clear non-additive or epistatic effects.Overall, the authors use a powerful combination of evolutionary analyses and biochemistry/molecular biology to identify novel functional regions of CD4. However, the author's conclusion that these motifs have coevolved to "finetune" the MHCII response are overstated and not supported by the functional data. Specifically, it is unclear what evidence the authors provide to support regulation of the ECD by intracellular or transmembrane regions. In figure 7, R426A, clasp, and TM mutants all decrease IL-2 on their own. The effects of these mutations in the background of the LL mutation seem to be simply additive, providing no evidence of interaction between these motifs (a la a network of coevolving residues).

Please note that we have removed the term “epistatic effects” from the manuscript. We previously used the term in a manner that we view as consistent with ‘epistatic’ interactions as described by Starr and Thornton (Starr and Thornton, 2016). They identify “*permissive epistatic interactions* [that] *made the residue tolerable in its native background, that restrictive epistatic mutations made it intolerable in the other, or both.*” In our system, the LL mutation leads to increased IL2 production. It is likely that the increased signaling that resulted in increased IL-2 would lower the fitness of an organism that acquired this mutation, although admittedly our current experimental setup does not allow us to test this directly. Individually, mutating the other motifs lowers the IL2 response. However, when combined with the LL mutation, the effects of mutating other motifs is indistinguishable from WT, indicating that these mutations counterbalance each other. To us, this suggests that the LL mutation, likely in combination with the non-polar patch with which it co-arose, restricted the evolutionary path of CD4 in mammals leading to the acquisition of the other regulatory motifs. Therefore, in our view, the LL and non-polar patch are examples of *permissive epistatic interactions* that allowed (and likely necessitated) the evolution of the other motifs identified in this study. We will explore this idea at some point in a future review where this speculative idea may be more appropriate.

The covariation could reflect a 'rheostat' wherein mutations in a stimulatory domain are compensated by mutations in an enhancing domain and vice versa, but there need be no regulation in the sense of a perturbation/input at one motif affects the function of another/allostery.

The comment that motifs with covarying residues could reflect a rheostat mediated by motifs with compensatory activity rather than allosteric interactions between the motifs is in line with our thinking. Indeed, we think it is more likely that residues in the nonpolar patch that stabilize TCR-CD3-pMHCII-CD4 assemblies are not allosterically interacting with the ICD helix but rather that interactions on the outside of the cell serve to dictate the spatial position of the CD4 ICD relative to the TCR-CD3 ICDs, and that the IKRLL motif of the ICD helix determines what the CD4 ICD interacts with and recruits to TCRCD3; however, we cannot rule out an allosteric effect. We did not intend to favor one or the other in the prior submission. Based on this helpful comment, we have tried in the revised Discussion section to be clear that experimentally determining the exact mechanistic basis for the function of the motifs we are studying individually, or in concert with each other, goes beyond the scope of the present study. In short, we’ve made observations that provide evidence of activity that has not been reported before, or even considered in play for CD4. These insights would have been hard to gain from less multi-disciplinary approaches and it is our hope that this study will serve as a hypothesis generator that opens up fertile ground for new discoveries.

To be suitable for a broad readership journal like eLife, the manuscript would benefit from a figure which shows the biological context (all the relevant molecules and the regions of interest in CD4). In addition, the diagrams included in the supplementary showing the mutations and their location within CD4 could be included in this or should be moved to the main figures. Similarly, some summary figure showing the mutations and their effects in a simple manner (up/down/wt) would be immensely useful to process the proposed interaction of mutations.

Thank you for the suggestions. We have now moved the structural model of CD4 to the main Figure 1 (Figure 1C), added Table 1 to describe the mutants, and added Table 2 to summarize the biochemical and functional impacts of the CD4 mutants relative to the WT. We were unclear on what exactly is being requested with regards to a figure showing the biological context, or where to call out such a figure in the text, as we read the suggestion as requesting the types of figures typically found in review articles. We have therefore refrained from making such a figure at this time. We can add one if needed. At issue is that we are unclear as to what would satisfy the requirement of “all relevant molecules” since there could be many. We did take care to overview many of the relevant molecules in the introduction (specifically, the third paragraph). Furthermore, the citations in that paragraph refer to several reviews with figures depicting molecules that are relevant to the current study. In particular, the reviews from the Weiss and Love Labs have what we think are nice illustrations that might satisfy the request (Courtney et al., 2018; Gaud et al., 2018). It is our hope that it is sufficient to reference these reviews for those within the broader community who are looking for more information in order to put our work in a broader context. Please let us know if you would prefer that we add a similar figure, and what level of detail is suitable.

The relevance of the functions measured is not clear – what do we know about the function of raft vs non-raft CD4? Most CD4 is in DSM/non-raft fraction. Why use CTxB and fractionation? Do they tell different things? What about when they don't agree? A clear setup of the why these assays have been chosen is crucial to understanding the relevance of the data.

First, we note that rafts are and have been extensively debated. We have adopted the definition outlined by Linda Pike as reported from the Keystone symposium on lipid rafts and cell function (Pike, 2006): “Membrane rafts are small (10–200 nm) heterogeneous, highly dynamic, sterol-and sphingolipid-enriched domains that compartmentalize cellular processes. Small rafts can sometimes be stabilized to form larger platforms through protein-protein and protein-lipid interactions.” Second, we note that it is now clear that cellular membranes are complex, with a variety of membrane rafts or islands with distinct lipid and protein compositions and that these play critical roles in cellular function (Lillemeier et al., 2010; Lillemeier et al., 2006).

What is known about the function of CD4 in membrane rafts is limited. One study reported a role in immunological synapse formation and another implicated an impact on intracellular signaling after antibody crosslinking (Balamuth et al., 2004; Fragoso et al., 2003). We have not seen studies that have related CD4 membrane raft localization to a downstream function, such as IL-2 production, in response to natural agonist pMHCII ligand as we have done here.

We used CTxB as a staining agent for GM1 (a ganglioside that is reported to concentrate in membrane rafts). Given that rafts are heterogenous, and sucrose gradient fractionation only segregates membranes and associated proteins into DRMs and DSMs, we used CTxB in combination with sucrose gradient fraction to ask if the motifs we are studying influences CD4 localization to distinct membrane domains within DRMs or DSMs. In other words, we interpret CTxB staining as providing additional information about the subdomains in which CD4 resides. Although this widely used approach is limited in the information it can provide, we think it is suitable for asking basic question about protein association with membrane subdomains. The results provide clear, albeit low resolution evidence that some of the motifs we are studying do influence membrane domain fractionation and thus justify future experimentation.

The language should be more precise regarding evolutionary inferences, with a clear delineation of what is a statistical test and what is an inference of those tests. For example:– The pairwise conservation/mutual information analysis detects statistical covariation which suggests coevolution. This analysis does not show coevolution.– They find statistical signatures of selection, not identify selection.– (ln 104-105) "how ancient and ongoing environmental challenges have influenced CD4" should read "find motifs that have been preserved".

To make the manuscript accessible to a broad audience we lost precision in how we described the evolutionary assays and their potential interpretation. We appreciate the reviewer pointing this out and have carefully reworded the text to clarify the distinction between test and interpretation.

The various monikers used for each motif are hard to keep track of. It would greatly help the reader to have a single term to apply to each motif in the text and figures (e.g. Palm and CV+C, ECD and Domain 3).

We appreciate that it can be hard to keep track of what mutations have been made and where they reside within CD4. To aid the reader, we have moved our model CD4 structure to the main Figure 1 and added Table 1, which lists the mutant name, location, the motif mutated, and the specific residues that are mutated for easy reference.

Reviewer #2 (Recommendations for the authors):"Enhancing and inhibitory motifs have coevolved to regulate CD4 activity" takes an evolutionary approach to investigate the mechanism(s) by which CD4 regulates TCR activation and downstream functional responses. The authors identify conserved motifs in the extracellular, transmembrane, and intracellular domains of CD4 that appear to regulate multiple aspects of its function, including its localization to lipid RAFTs, its capacity to interact with Lck, and its ability to promote IL-2 production. Notably, one of these motifs, comprising a helix just N-terminal to the cysteine clasp responsible for interaction with Lck, has an inhibitory effect on TCR signaling, and it seems to have coevolved in eutherians together with a motif in extracellular domain 3 that promotes T cell activation. Collectively, these results suggest that the regulation of CD4 activity has been finely tuned during evolution such that the acquisition of activity-promoting mutations is balanced by the emergence of inhibitory regions. Although this idea is interesting, it should be accompanied by more in-depth mechanistic work to demonstrate exactly how specific parts of CD4 control TCR activation. The notion that motifs within CD4 have coevolved is not particularly unexpected. These evolutionary relationships should be connected to mechanistic and functional insights into how the molecule works.Regarding: “Although this idea is interesting, it should be accompanied by more indepth mechanistic work to demonstrate exactly how specific parts of CD4 control TCR activation.”

In consideration of this comment we have worked to rule in or out the most obvious mechanism, based on the TCR signaling paradigm and related models, for how the CD4 mutants with the most dramatic Lck and IL-2 phenotypes (Clasp, TP, and LL) influence TCR-CD3 signal initiation (Glaichenhaus et al., 1991; Rudd, 2021; Stepanek et al., 2014). The TCR signaling paradigm posits that CD4 contributes to TCR-CD3 signal initiation by recruiting Lck, via interactions at the CQC clasp, to phosphorylate the CD3 ITAMs when TCR-CD3 and CD4 both engage pMHCII. This model predicts that our Clasp and LL mutants, both of which reduce CD4-Lck association, should lead to reduced phosphorylation of the CD3z ITAMs. In contrast, because our TP mutant increases CD4-Lck association, it should lead to increased CD3z ITAM phosphorylation. We also analyzed phosphorylation of another substrate of Lck (Zap70), as well as an indirect downstream target (Plcg1), to gain deeper insights into how the Clasp, TP, and LL mutants impact TCR-CD3 proximal signaling events upstream of IL-2 production.

The results are presented in Figure 6. To summarize: 1) our Clasp mutant does not impact CD3z ITAM phosphorylation but does reduce Zap70 and Plcg1 phosphorylation compared to WT CD4; 2) our TP mutant reduces phosphorylation of all three molecules compared to WT CD4; 3) our LL mutant does not have a discernable impact on the phosphorylation of any of these signaling intermediates compared to WT CD4. Our interpretation of these results, and their implications, are detailed in the Discussion (for quick reference: the Clasp mutant (CQC motif) reduces CD4-Lck interactions and IL-2 production; the TP mutant (GGXXG plus CV+C motifs) increases CD4-Lck interactions, localizes CD4-Lck pairs to DSMs, and reduces IL-2; the LL mutant (IKRLL motif) decreases CD4-Lck interactions and increases IL-2 production.)

We will note here that the signaling events that originate at TCR-CD3-pMHCII-CD4 assemblies split into multiple signaling cascades, involving multiple intermediates and second messengers, to direct nuclear localization of multiple transcription factors that together drive IL-2 expression (our endpoint readout for pMHCII-specific signaling) (Courtney et al., 2018; Gaud et al., 2018; Malissen and Bongrand, 2015). We could form several hypothesize to explain how, for example, the IKRLL motif (mutated with our LL mutant) regulates TCR-CD3 signaling but we cannot test them all in one study to conclusively demonstrate how this one mutant functions. We think the most likely explanation for the inhibitory activity of the IKRLL motif is that it is mediated by an unidentified interacting partner. Finding this putative partner will require a screen and extensive characterization. Therefore, while we agree that further mechanistic analysis of the motifs described here is warranted, we ask the reviewer to please consider that demonstrating exactly how specific parts of CD4 control pMHCII-specific signaling could take years and numerous additional datasets. For this reason, we hope the reviewer will agree that doing so goes beyond the scope of the current study but that our body of data make a compelling case that further study is warranted.

Regarding: “The notion that motifs within CD4 have coevolved is not particularly unexpected.”

To our knowledge, the question has not been asked or investigated for CD4 or other components of multi-module activating immune receptors.

Regarding: “These evolutionary relationships should be connected to mechanistic and functional insights into how the molecule works.”

Our study was designed to describe evolutionary, mechanistic, and functional relationships by relating how the distinct motifs identified by our evolutionary analysis influence membrane domain localization, CD4 association with Lck, and IL-2 production in response to pMHCII. We chose membrane domain localization and Lck association because both have been proposed to regulate CD4 function, and we chose IL-2 production as an endpoint readout for differences in pMHCII-specific signaling (Fragoso et al., 2003; Glaichenhaus et al., 1991; Stepanek et al., 2014). We acknowledge that differences in IL-2 production can tell us that there is a difference in signaling between a particular mutant of CD4 and the WT molecule, but not where and in what pathway that defect lies. As detailed above, we have added analysis of TCR-CD3 proximal signaling events. We think that our results support the conclusion that CD4 function is more complex than can be explained by the TCR signaling paradigm and related models. We appreciate Reviewer #2’s view that more mechanistic work is needed and trust that, once our study is available to the field, the current study will help fuel interest in this topic.

1) Just because motifs have coevolved and their mutations have additive effects doesn't necessarily mean that they are functionally coupled. Mechanistic insight will require a more in-depth analysis of membrane proximal signaling events and the activation status of downstream pathways (e.g. Erk, Ca, NFKB).

We appreciate this comment. As with any study, we have put forth an interpretation of our data that we think is most consistent with all of the results, and other published work. We acknowledge that we cannot know with certainty why particular residues or motifs have coevolved. That being written, there is a growing body of evidence that points to a functional relationship between residues that coevolve, even if they are located at distant sites.

We have removed the term “functionally coupled” from the revision in consideration that there may be a difference in perspective regarding how it should be used or interpreted. We now use the term “counterbalance” to describe the neutral phenotype (equal to WT) we observe when we combine a motif that reduces IL-2 with one that enhances IL-2.

As discussed above, we have added additional mechanistic insights about our Clasp and TP mutants. We agree that more is needed and think that there is a long road ahead in understanding the function of the ICD helix, and IKRLL motif therein, as it is clearly a multifunctional hub with which other residues have covaried over evolutionary time. For the reasons outlined above, we think that providing the exact mechanistic details of how different motifs interact functionally requires that we first understand the function of the ICD helix, and IKRLL motif therein. We think that doing so goes beyond the scope of this study.

2) The idea that the role of CD4 is to control the colocalization of the TCR and Lck to RAFTs is quite interesting, and it should be tested. Could RAFT localization be modulated independently of CD4, and would this reverse the effects of the relevant CD4 mutations?

We agree that this is an interesting question and think that interrogating it properly would require a separate study. There are real technical challenges to doing this well.

3) It is difficult to get a sense of what the fractionation and lipid association results really mean in terms of membrane organization (e.g. Figure 2). CTxB binding in the context of a bead-based IP is particularly unphysiological. A positive control would be helpful here. What would an established disruption of RAFT localization look like in this assay (e.g. a mutation in the Lck N-terminal domain).

While membrane domains and their composition are clearly important for several biological processes, they are challenging to study. We used a classic, well-accepted biochemical technique (sucrose gradient fractionation) and a quantitative flow-based IP readout (basically a bead-based ELISA) to evaluate how the motifs we are studying impact segregation of CD4 molecules or CD4-Lck pairs into DRMs (membrane rafts) or DSMs (non-rafts)(Fragoso et al., 2003; Glassman et al., 2016; Parrish et al., 2016; Schrum et al., 2007). One reason for using this approach is that the impact of mutating some of the motifs studied here on sucrose gradient fractionation has already been published and thus those studies provide us with a benchmark for comparison [i.e. the CQC clasp motif, the CV+C palmitoylation motif, and the HXXR motif (shown in original submission but removed from the revision) (Fragoso et al., 2003; Popik and Alce, 2004)]. Those studies showed in single representative Western blots that mutating the CQC, CV+C, and HXXR motifs reduced CD4 localization to DRMs. Mutating these motifs yields similar results in our analysis. We hope that this comparison addresses the positive control comment.

We stained with CTxB because it binds the ganglioside GM1 that is reported to be enriched in membrane rafts. We agree that CTxB binding of bead-based IP is not physiological, although Western blot analysis of sucrose fractionated cell lysates is also not physiological. We looked at CTxB staining because we thought it might provide further insights into CD4 localization in membrane subdomains within DRMs and DSMs. We think the finding that our Clasp and Palm mutants have very different CTxB staining within the DRMs suggests that these two motifs control localization to distinct subdomains within DRMs for those CD4 molecules still localized in DRMs. We also think it is interesting that our TMD and Palm mutants have the same CTxB staining in DRMs as WT CD4 and that the TP mutant (TMD+Palm) has less CTxB staining that the Palm mutant. These data suggest that together, the GGXXG and CV+C motifs impact membrane domain, or subdomain, localization differently than either one individually. We do not know exactly what this means about membrane organization as the approach is admittedly low-resolution and our interpretation is limited. Nevertheless, we included the data in the manuscript because we think that some people in the field might find it interesting, useful, or otherwise hypothesis generating. We can remove the CTxB analysis from the manuscript if that is preferred.

4) Based on the presented data, one might conclude that mutation of the ICD helix enhances T cell responses simply by inhibiting CD4 internalization, thereby maintaining the density of CD4 on the cell surface. Indeed, all of the mutations that enhance IL-2 responses appear to result in higher surface expression. Do the authors favor this hypothesis, or can it be ruled out?

We have added additional data to address this point. Specifically, we have analyzed additional ICD helix mutants in Figure 4 to understand how serines flanking the IKRLL motif influence the inhibitory function of the IKRLL motif. Endocytosis of all of the ICD helix mutants is impaired, yet they have distinct IL-2 phenotypes suggesting that IL-2

production is more heavily influenced by the ICDs than expression levels. In addition, we included data with a C-terminally truncated CD4 molecule, termed CD4-T1 in accordance with previous labeling, for a mutant that is truncated immediately after the CV+C motif and thus lacks the remaining ICD, including the helix (see Table1 and Figure 4 —figure supplement 5) (Glaichenhaus et al., 1991). CD4-T1 expresses at equivalent levels to the LL mutant, and fails to endocytose, but does not direct increased IL-2 production relative to the WT the way the LL mutant does. Instead, it directs slightly reduced IL-2 production. These data clearly indicate that the differences in IL-2 production between the CD4-T1 and LL mutants are due to differences in their ICDs and not their expression levels at the initiation or throughout the co-culture with APCs.

Of additional note, we have moved data concerning the H and HC mutants to Figure 3. In revising the manuscript, we realized that the cell lines used for the flow cytometry and sucrose data were not the same as those used in the IL-2 (not generated on the same day). We have now adjusted the data such that all data is from a single set of cell lines generated at the same time. Specifically, we updated the flow cytometry and sucrose data because the cell lines shown throughout the revised figure was more closely matched in CD4 expression, which is why they had originally been chosen for the IL-2 data. Now, for all figures in the manuscript, the flow data, sucrose gradient data, and IL2 data are all from the same set of lines wherein the WT and mutants being compared were generated at the same time.

5) Given the evolutionary focus of this study, it would have been more interesting if the authors had actually used sequences from non-eutherian CD4s, as opposed to Ala and Gly substitutions. Was this attempted?

The goal of this study was to better understand CD4 in general, and eutherian CD4 in particular. We focused specifically on mouse and human CD4 since the former is an extensively-used model for the latter, which has obvious implications for immunotherapeutic engineering and human health. This is in accordance with our NIH funded project. Given our goal, we decided to take a more conventional structure function approach and use mutations that would disrupt the chemical nature of the motifs under investigation.

We think that replacing eutherian sequences with those from non-eutherian CD4 would also be incredibly interesting; however, such experiments would be performed in response to a different question with a different end-goal in mind. We also think this approach may be harder to interpret because the non-eutherian CD4 homologous evolved along a different trajectory, in potentially different cellular environments (e.g. temperature, membrane composition). We do not know how to predict how non eutherian motifs would behave in the background of mouse CD4 in mouse cells. We also think there may be many caveats associated with interpreting such data. Overall, we think this is a different question that would be worthy of a separate study.

6) It is hypothesized that the importance of the GGXXG motif may depend on the local cholesterol concentration of the membrane domain in which it resides. Does changing membrane cholesterol modulate the effects of mutations in the GGXXG motif?

Changing membrane cholesterol levels or interfering in other ways is expected to have effects beyond CD4, making any results challenging to interpret (Chen et al., 2022; Swamy et al., 2016; Wang et al., 2016). Therefore, we thought that speculating about the interplay between cholesterol concentrations and the GGXXG motif was more appropriate for the Discussion.

7) The authors should confirm that the CV+C mutation actually has the intended consequence of altering CD4 palmitoylation.

Because previous publications have established that the cysteines of the CD4 CV+C motif are plamitoylated, and other studies have shown that similar motifs are also palmitoylated in other proteins, we consider this to be well established in the field (Arcaro et al., 2000; Crise and Rose, 1992; Fragoso et al., 2003; Ladygina et al., 2011).

Reviewer #3 (Recommendations for the authors):In this manuscript, Lee et al., the authors interrogate the function of CD4 from an evolutionary perspective. Adaptive immunity within the jawed vertebrates is believed to have arisen as a consequence of the "Immunological Big Bang". This event marks the evolutionary origin of both T and B cell receptors, as well as many of the molecules that mediate their signaling (e.g., the TCR co-receptors CD4 and CD8, as well as the MHC I and II proteins). Although orthologs of TCRs, CD4, and MHCII are broadly conserved throughout jawed vertebrates, these proteins have not yet been identified outside of this lineage. Notably, all three molecules first appear within the cartilaginous fishes, suggesting that their functions may be tightly linked. However, most studies on TCR function focus specifically on TCRs interacting directly with either CD4 or MHCII. In this study, the authors draw on evolutionary analyses to identify broadly conserved sequence motifs that may be involved in direct interactions between CD4 and MHCII to fine-tune the strength of TCR signaling. They complement these computational analyses with biochemical studies to support their hypothesis.The primary strength of this manuscript lies in the author’s integrative approach; their evolutionary hypotheses are considerably strengthened by the findings from the experimental techniques. However, in its current form, the manuscript is very difficult to follow for a non-specialist in TCR signaling. The authors use a considerable amount of jargon and abbreviations. Consequently, the logic underlying their experiments is not always clear. Thus, while their biological findings may be of interest to researchers interested in the specifics of TCR signaling (particularly from the standpoints of engineering biomimetics and clinically manipulating TCR signaling strength), it is difficult to extend the significance of these findings more broadly.

See specific comments below.

As mentioned above, the manuscript contains a nearly impenetrable amount of abbreviations and jargon. I suggest that the authors have the manuscript revised by someone outside the field who is familiar with the work to address this problem. This will make the manuscript more considerably more accessible to a broad range of biologists.

We understand and agree that field-specific nomenclature can, at times, feel like a foreign language to those outside of the field. Indeed, as an interdisciplinary collaboration, our team has had to spend time and effort learning each other’s language. This has been challenging, but also fun. We appreciate the suggestion and will note that we did have colleagues who understand the evolutionary analysis but not the immunology, and vice versa, read the manuscript and provide comments. Their suggestions were incorporated into the initial submission as we anticipated that we were likely to get one or more comments such as this and can assure the reviewer we have tried to make the manuscript accessible. We will note that our attempts backfired somewhat as Reviewer #1 took issue with the precision of our language regarding our evolutionary analysis which was intended to make the work more accessible.

We ask the reviewer to consider that we are constrained in how we relate our work to the reader. The immunological terms, including names of the cell types and proteins, must be kept consistent with the field in order for people inside and outside the field to be able to relate our work to that of others in the field. And, as called out by Reviewer #1, we must also use precision when discussing the evolutionary aspects of the work. In short, we think that we are generally obliged to stick to convention when discussing key aspects of this work. With regards to other nomenclature, or jargon, we have tried to use standard terms. For example, we used ECD, TMD, and ICD to refer to the extracellular, transmembrane, and intracellular domains of a protein as is common in biochemistry and molecular and cellular biology because these abbreviations should be broadly accessible. Similarly, referring to the detergent resistant and detergent soluble membrane fractions of sucrose gradients as DRMs and DSMs is conventional. We would be happy to use the long form instead of the abbreviations if Reviewer #3 and the editor think it would be more appropriate and we are not constrained by space.

Finally, we acknowledge that the mutant names may be a problem as we sometimes have a hard time keeping track of all the mutants. For this reason, we chose to name our mutants in a way that relates to their location or particular motif. For example, we found that calling the CQC clasp motif mutant “Clasp” and our CV+C palmitoylation motif mutant “Palm” was easier for us to keep track of than using a residue-based naming system such as C444S+C446S and C418S+C421S, respectively. We have now added Table #1 as a quick reference for the mutant names, locations, residue number being changed, and the nature of the mutation. We hope that this helps.

Two additional points that should be addressed:1. On line 55, the authors state that "[in sharks]… an orthologous gene encoding CD4appears to be absent, as are genes for proteins associated with CD4^+^ 56 T cell helper (Th) or regulatory (Treg) functions (e.g. FoxP3 and Rorc) (Venkatesh et al., 2014)" Although this was the primary finding described in the original elephant shark genome paper, subsequent analysis of this genome (JM Dijkstra, Nature 511, 2014) revealed that the CD4^+^ T cell lineage is likely present within cartilaginous fish.

We appreciate this comment and have removed this mention of the Venkatesh paper from our manuscript due to the controversial nature of their finding.

2. The manuscript relies heavily on evolutionary analysis of CD4 molecules collected from jawed vertebrates. However, the authors do not adequately describe if/how they verified the homology of these molecules. The authors state that "The criteria used for including orthologous CD4 sequences in our analyses were that they have a domain structure consisting of four extracellular Ig domains followed by a TMD and a C-terminal ICD". Many proteins have convergently evolved this domain architecture, which makes this criteria insufficient to assign orthology. A phylogenetic approach is required to confidently determine whether or not the sequences are truly orthologous to CD4. Additionally, all accession numbers and/or aligned sequences should be made freely available in the supplemental material.

As the reviewer points out, it is hard to demonstrate that proteins are orthologous. Our dataset is based on reciprocal blast-based searches of the NCBI database. Sequences were aligned and used to construct phylogenetic trees. Sequences that either did not have a canonical CD4 structure or contained large indels were removed from the analysis. Likewise, we assumed that the evolution of CD4 should reflect that of the host species. Sequences that did not cluster as expected were removed from the dataset. The final dataset, the aligned sequences, and phylogenetic trees are available from Dryad (https://doi.org/10.5061/dryad.59zw3r26z).

References cited in our response to reviewers:

Arcaro, A., Gregoire, C., Boucheron, N., Stotz, S., Palmer, E., Malissen, B., and Luescher, I.F. (2000). Essential role of CD8 palmitoylation in CD8 coreceptor function. J Immunol *165*, 2068-2076.

Balamuth, F., Brogdon, J.L., and Bottomly, K. (2004). CD4 raft association and signaling regulate molecular clustering at the immunological synapse site. J Immunol *172*, 58875892.

Chen, Y., Zhu, Y., Li, X., Gao, W., Zhen, Z., Dong, Huang, B., Ma, Z., Zhang, A., Song, X.*, et al.* (2022). Cholesterol inhibits TCR signaling by directly restricting TCR-CD3 core tunnel motility. Mol Cell.

Courtney, A.H., Lo, W.L., and Weiss, A. (2018). TCR Signaling: Mechanisms of Initiation and Propagation. Trends in biochemical sciences *43*, 108-123.

Crise, B., and Rose, J.K. (1992). Identification of palmitoylation sites on CD4, the human immunodeficiency virus receptor. J Biol Chem *267*, 13593-13597.

Davey, N.E., Cyert, M.S., and Moses, A.M. (2015). Short linear motifs – ex nihilo evolution of protein regulation. Cell Commun Signal *13*, 43.

Fragoso, R., Ren, D., Zhang, X., Su, M.W., Burakoff, S.J., and Jin, Y.J. (2003). Lipid raft distribution of CD4 depends on its palmitoylation and association with Lck, and

evidence for CD4-induced lipid raft aggregation as an additional mechanism to enhance CD3 signaling. J Immunol *170*, 913-921.

Gaud, G., Lesourne, R., and Love, P.E. (2018). Regulatory mechanisms in T cell receptor signalling. Nat Rev Immunol *18*, 485-497.

Glaichenhaus, N., Shastri, N., Littman, D.R., and Turner, J.M. (1991). Requirement for association of p56lck with CD4 in antigen-specific signal transduction in T cells. Cell *64*, 511-520.

Glassman, C.R., Parrish, H.L., Deshpande, N.R., and Kuhns, M.S. (2016). The CD4 and CD3deltaepsilon Cytosolic Juxtamembrane Regions Are Proximal within a Compact TCR-CD3-pMHC-CD4 Macrocomplex. J Immunol *196*, 4713-4722.

Kim, P.W., Sun, Z.Y., Blacklow, S.C., Wagner, G., and Eck, M.J. (2003). A zinc clasp structure tethers Lck to T cell coreceptors CD4 and CD8. Science *301*, 1725-1728.

Kobayashi, S., Thelin, M.A., Parrish, H.L., Deshpande, N.R., Lee, M.S., Karimzadeh, A., Niewczas, M.A., Serwold, T., and Kuhns, M.S. (2020). A biomimetic five-module chimeric antigen receptor ((5M)CAR) designed to target and eliminate antigen-specific T cells. Proc Natl Acad Sci U S A *117*, 28950-28959.

Ladygina, N., Martin, B.R., and Altman, A. (2011). Dynamic palmitoylation and the role of DHHC proteins in T cell activation and anergy. Adv Immunol *109*, 1-44.

Lillemeier, B.F., Mortelmaier, M.A., Forstner, M.B., Huppa, J.B., Groves, J.T., and Davis, M.M. (2010). TCR and Lat are expressed on separate protein islands on T cell membranes and concatenate during activation. Nature immunology *11*, 90-96.

Lillemeier, B.F., Pfeiffer, J.R., Surviladze, Z., Wilson, B.S., and Davis, M.M. (2006). Plasma membrane-associated proteins are clustered into islands attached to the cytoskeleton. Proc Natl Acad Sci U S A *103*, 18992-18997.

Malissen, B., and Bongrand, P. (2015). Early T cell activation: integrating biochemical, structural, and biophysical cues. Annu Rev Immunol *33*, 539-561.

Parrish, H.L., Deshpande, N.R., Vasic, J., and Kuhns, M.S. (2016). Functional evidence for TCR-intrinsic specificity for MHCII. Proc Natl Acad Sci U S A.

Parrish, H.L., Glassman, C.R., Keenen, M.M., Deshpande, N.R., Bronnimann, M.P., and

Kuhns, M.S. (2015). A Transmembrane Domain GGxxG Motif in CD4 Contributes to Its Lck-Independent Function but Does Not Mediate CD4 Dimerization. PLoS One *10*, e0132333.

Pike, L.J. (2006). Rafts defined: a report on the Keystone Symposium on Lipid Rafts and Cell Function. J Lipid Res *47*, 1597-1598.

Popik, W., and Alce, T.M. (2004). CD4 receptor localized to non-raft membrane microdomains supports HIV-1 entry. Identification of a novel raft localization marker in CD4. J Biol Chem *279*, 704-712.

Rudd, C.E. (2021). How the Discovery of the CD4/CD8-p56(lck) Complexes Changed Immunology and Immunotherapy. Front Cell Dev Biol *9*, 626095.

Schrum, A.G., Gil, D., Dopfer, E.P., Wiest, D.L., Turka, L.A., Schamel, W.W., and Palmer, E. (2007). High-sensitivity detection and quantitative analysis of native proteinprotein interactions and multiprotein complexes by flow cytometry. Sci STKE *2007*, pl2.

Starr, T.N., and Thornton, J.W. (2016). Epistasis in protein evolution. Protein Sci *25*, 1204-1218.

Stepanek, O., Prabhakar, A.S., Osswald, C., King, C.G., Bulek, A., Naeher, D., BeaufilsHugot, M., Abanto, M.L., Galati, V., Hausmann, B.*, et al.* (2014). Coreceptor scanning by the T cell receptor provides a mechanism for T cell tolerance. Cell *159*, 333-345.

Swamy, M., Beck-Garcia, K., Beck-Garcia, E., Hartl, F.A., Morath, A., Yousefi, O.S., Dopfer, E.P., Molnar, E., Schulze, A.K., Blanco, R.*, et al.* (2016). A Cholesterol-Based Allostery Model of T Cell Receptor Phosphorylation. Immunity *44*, 1091-1101.

Wang, F., Beck-Garcia, K., Zorzin, C., Schamel, W.W., and Davis, M.M. (2016). Inhibition of T cell receptor signaling by cholesterol sulfate, a naturally occurring derivative of membrane cholesterol. Nat Immunol *17*, 844-850.

[Editors’ note: what follows is the authors’ response to the second round of review.]

The manuscript has been improved but there are some remaining issues that need to be addressed, as outlined below:1) The reviewers feel that the new signaling data are conceptually the right experiments but may be technically limited by overstimulating T cells and sparsely sampling antigen concentrations in some assays. As discussed by Reviewer 3, despite the sparse dose response applied in the signaling experiments, the manuscript presents the conclusions as if there can be only one interpretation of the results.For example, whether or not the Clasp plays a positive or negative role in TCR signaling is not settled science (which is why your manuscript is interesting). The reviewers don't feel like the added experiments completely resolve these questions, so we request that your either add additional experiments or appropriately temper the presentation and discussion of your conclusions to acknowledge the remaining gaps in our understanding of this important topic.

We appreciate this comment and have taken your suggestions to modify the manuscript. In brief:

Please note that we have now also added analysis of two additional independently generated WT and Clasp mutant cell line pairs to our proximal signaling analysis in Figure 6A for a total of four independent generated lines. While we found no difference in the pCD3z MFI compared to the WT for the two independently generated cell lines we originally tested, one of the two additional cell line showed pCD3z MFI that was ~83% of WT. The other line showed a very small reduction in pCD3z MFI (~92% of WT), which is nevertheless statistically significant. The key point remains that we do not see an absence of pCD3z MFI with the Clasp mutant, nor do we see a consistent reduction in pCD3z MFI as is expected based on the prevailing paradigm.

Thank you. We followed this suggestion and asked a colleague, Dr. Benjamin Renquist who works on obesity in mice, to read the manuscript and provide suggestions on how to make the manuscript more accessible. We greatly appreciate the gift of his time and have worked to incorporate his suggestions into the manuscript.

We understand the difficulty of crafting a narrative that is accessible to all readers but still respects the conventions and standards of both immunology and molecular evolution. While we appreciate the revisions made to improve the readability of the manuscript, we respectfully request that you have a colleague outside the evolution and immunology read the manuscript and highlight sections or terminology or acronyms that could be modified with the goal of increasing the accessibility of the narrative. The extent and substance of these revisions are left to your discretion.Reviewer #2 (Recommendations for the authors):In their manuscript, "Enhancing and inhibitory motifs regulate CD4 activity", Lee et al., employ an evolutionary approach to identify specific residues within CD4 that impact TCR signaling. CD4 is a transmembrane protein composed of four immunoglobulin domains that mediates T cell signaling in two ways: first, through direct interactions with MHC II on antigen presenting cells; and second, by facilitating intracellular signaling by interacting with the Src kinase, Lck. Traditionally, the researchers have focused on the TCR-MHC II interactions as the key determinants of T cell signaling. However, recent evidence suggests that CD4 may also regulate the outcome of T cell activation. In this study, the authors use phylogenetic analysis as a relatively unbiased means of defining residues within CD4 that are important for signaling. This is predicated on the assumption that protein evolution is constrained by function; residues that directly mediate signaling are less likely to mutate over time. CD4 is a central component of the adaptive immune system and its function is limited to this context. Accordingly, orthologs of CD4 have only been identified within vertebrates.In this study, the authors use computational analyses to characterize the evolution of CD4 within the jawed vertebrates as a foundation for functional experiments. Overall, the manuscript is well-written and the conclusions are supported by the data shown. This work could potentially be of interest to a broad group of researchers, including immunologists interested in adaptive immunity and comparative immunologists as well as cell biologists who are focused on transmembrane cell signaling proteins.My primary concern with the manuscript is that, in its current form, it is difficult to interpret by a non-specialist in TCR signaling. The manuscript contains a considerable amount of jargon, abbreviations and specific details without the necessary context. I would suggest having the manuscript read by someone who is not familiar with the work and can highlight the more challenging sections.

We understand the comment and thank you for the suggestion. We did have another colleague, Dr. Benjamin Renquist who studies obesity in mouse models, read the paper to provide additional critical feedback. He also indicated that the numerous acronyms and sheer number of mutants made for a tough read. He found that he could navigate the manuscript with the aid of Figure 1C and Table 1, suggested by the reviewers (thank you), and he suggested as with Reviewer #3 that we should take the space to spell out key acronyms. We have now done this. He also suggested adding an acronym key to Table 2, which we have now done. In addition, following this line of thinking, we have added acronym keys to the figure legends.

In a similar vein, the figures are quite complex. The authors might consider simplifying to emphasize the main points.

We also appreciate this comment but are unsure how to further modify them beyond what we have already done. As noted above, we did add acronym keys to the figure legends.

Specific points to address are outlined below.1. In the Introduction, the authors refer to the "immunological 'Big Bang' that gave rise to RAG-based antigen receptor gene rearrangement in jawed vertebrates…". This discounts several important advances in the evolution of adaptive immunity, including the characterization of the VLR-based adaptive immune system that evolved in parallel within the jawless vertebrates as well as the discovery of orthologs of the RAG1/2 and CDA enzymes within invertebrate deuterostomes. These findings, which are described and synthesized in Flajnik, 2014 (doi:10.1016/j.cub.2014.09.070), will provide important evolutionary context for this manuscript.

We appreciate the comments and suggestion. We think these are all important evolutionary developments in adaptive immunity that serve as a nice introduction to the paper from “30,000ft” before zooming in specifically on the evolution of CD4 and its role in T cell biology. We have now cited the Flajnik paper to point interested readers to a key resource for a broader sense of these important issues. We now take more care to be clear to state that this paper is focused on adaptive immunity in jawed vertebrates.

2. Line 69. The authors describe MHC-TCR signaling as mediated by "5 distinct modules", but it is not clear from the subsequent text what these modules are. This should be clarified.

We appreciate this suggestion and have now more carefully described each module.

3. Although the authors describe MHC-TCR signaling in great detail, the structure of CD4 is not described. This would be particularly useful to readers who are not specialists in T cell biology and might include a figure showing the four Ig domains, the transmembrane region, and the intracellular domain. The abbreviations ECD, TMD, and ICD should be clearly defined early in the text.

Figure 1C shows a model of CD4 with the elements highlighted as suggested. We have also added an acronym key to the figure legend to aid the reader in understanding the figure.

4. Line 115. The authors should briefly mention in the text which CD4 orthologs were used. For example, "X CD4 orthologs were analyzed from Y species, which included the vertebrate lineages …". This should be further elaborated in the methods section. The authors state (line 682) that "… available CD4 orthologs were downloaded from Genbank."

We have updated to text to read:

We performed multiple analyses of available vertebrate CD4 ortholog sequences (n=99 distinct sequences), representing ~435 million years of evolution, to understand how ancient and ongoing environmental challenges have influenced CD4. The analyzed sequences represent fish, reptiles (including birds), marsupials, and placental mammals. Details related to ortholog selection are outlined in Materials and methods. All sequences and files are available through the DataDryad repository associated with this manuscript.

More details are necessary. There are many transmembrane proteins that consist of four Ig domains; only a subset of these have been correctly annotated as CD4. The authors should specifically describe their inclusion/exclusion criteria (i.e., define the algorithms and parameters used and the "etc" on line 685).

Available CD4 orthologs were identified through reciprocal blast-based searches and downloaded from GenBank. BLAST may not only identify orthologs so additional criteria were used for including putative orthologous CD4 sequences in our analyses: presence of a domain structure consisting of four extracellular Ig domains followed by a transmembrane domain and a C-terminal intracellular domain, including the presence of the Lck binding clasp (CxC). Sequences that were shorter, contained frameshift mutations, or displayed high sequence variability were excluded from the analysis.

Accession numbers for all these sequences should be included in the manuscript.

Accession numbers are available through the DataDryad repository associated with this manuscript.

5. Although most mammals have a single ortholog of CD4, due to whole genome duplications, many teleost genomes contain two paralogs (e.g., doi:10.4049/jimmunol.1600222). Were these included in the analysis?

Based on the criteria described above we filtered out potential duplicates.

Conversely, the authors cite Venkatesh et al., 2014 as evidence of the absence of CD4 in sharks. This manuscript was followed by a comment (doi:10.1038/nature13446) that identified many of the "missing" components of adaptive immunity from the elephant shark genome sequence. It would be wise to double-check these findings.

We removed the statement in question and reference to the Venkatesh paper.

6. Line 131. It is not clear how the distribution of residues subject to purifying selection suggests that "mutation putative linear motifs within the ICR is selected against". This should be clarified or justified.

The data demonstrates that 45% of the residues in the intracellular domain of CD4 are under purifying selection. We interpret this to demonstrate that mutating these residues that are primarily located in unstructured protein regions negatively affect fitness and are selected against.

Reviewer #3 (Recommendations for the authors):This paper takes an evolutionary approach to investigate the mechanism(s) by which CD4 regulates TCR activation and downstream functional responses. The authors identify conserved motifs in the extracellular, transmembrane, and intracellular domains of CD4 that appear to regulate multiple aspects of its function, including its localization to lipid RAFTs, its capacity to interact with Lck, and its ability to elicit membrane proximal signaling and promote IL-2 production. Notably, one of these motifs, comprising a helix just N-terminal to the cysteine clasp responsible for interaction with Lck, has an inhibitory effect on TCR signaling, and it seems to have coevolved in eutherians together with a motif in extracellular domain 3 that promotes T cell activation. Collectively, these results suggest that the regulation of CD4 activity has been finely tuned during evolution such that the acquisition of activity-promoting mutations is balanced by the emergence of inhibitory regions.The revisions and new data have improved this manuscript, although I still have some concerns.1) The manuscript remains a difficult read. I'm not sure how best to deal with this. The acronyms get a bit overwhelming. Perhaps it would be better to remove some of the less used acronyms (e.g. MRCA) and just write the words out in full?

We understand and appreciate both the feedback and suggestion. We have written out the acronyms in the manuscript.

2) The new signaling data (Figure 6) are a welcome addition to the manuscript, but they are rather sparse. In particular, I am worried that the authors may be missing important phenotypes by overstimulating the TCR expressing cells. A dose response should be performed for each signaling readout. It's possible, for instance, that at a lower antigen dose they would observe more substantial differences in CD3z phosphorylation in Figure 6A. This would be consistent with the idea that CD4-Lck interaction matters more when antigenic peptide is limiting.

We appreciate the comment and agree that our Clasp mutation might impair CD3z phosphorylation at low levels of agonist MCC, particularly at the levels where CD4 has been shown to be critical for calcium mobilization as a readout of proximal signaling events (<20). However, formally testing this in a convincing manner would be very challenging and would take a good deal of time. We have therefore addressed the concern as follows:

Please know that we considered the potential pitfall of overstimulation, and other technical issues, during internal deliberations about how best to study the influence of CD4 on proximal signaling events. After several pilot experiments, we settled on an approach that we consider to be robust, well-controlled, and interpretable within the parameters tested.

Key considerations in designing our experiments were that we wanted to: 1) synchronize TCR engagement on our WT and mutant cell lines; 2) use agonist pMHCII to engage TCR-CD3 and CD4 with their physiological ligands to avoid non-physiological signaling; 3) generate detectable levels of signal to be able to confidently distinguish signal from noise for data interpretation; and, 4) distinguish between differences in the frequency of responding cells and differences in the intensity of responses among responding cells, which is something that cannot be achieved with bulk cell signaling assays such as Western blots.

Antibody crosslinking (e.g. anti-CD3) is typically used in T cell signaling studies to activate cells. This approach provides reasonable control over the timing of signaling initiation, such that many cells within a sample can be activated with relative synchrony due to the high levels of antibody used to ensure robust signal. We were concerned that using anti-CD3/anti-CD4 crosslinking to synchronize activation of our sample populations would stimulate our cells in a non-physiological, and potentially overstimulating manner due to many factors including drastically slowed kinetics of engagement relative to TCR-CD3 and CD4 engagement of pMHCII on APCs (Glassman et al., 2018). We therefore reasoned that, with such an approach, any differences we might observe between the WT and mutant cells may not faithfully mirror any differences that result from engagement of agonist pMHCII, making it hard to confidently interpret the results. Using APCs expressing high densities of agonist pMHCII ensures that when the T cells and APCs are spun together and shifted to 37ºC to initiate signaling, that TCRs and CD4 should rapidly find their ligands to initiate signaling in sufficient numbers to allow for detectable signals with current approaches.

We are concerned that experiments using APCs with low doses of agonist pMHCII would be challenging for the following reasons: